



# Description of a formaldehyde retrieval algorithm for the Geostationary Environment Monitoring Spectrometer (GEMS)

Hyeong-Ahn Kwon[1], Rokjin J. Park[1], Gonzalo González Abad[2], Kelly Chance[2], Thomas P. Kurosu[3], Jhoon Kim[4], Isabelle De Smedt[5], Michel Van Roozendael[5], Enno Peters[6,*], and John Burrows[6]

[1]School of Earth and Environmental Science, Seoul National University, Seoul, Republic of Korea
[2]Atomic and Molecular Physics Division, Harvard-Smithsonian Center for Astrophysics, Cambridge, Massachusetts, USA
[3]Earth Science, Jet Propulsion Laboratory, Pasadena, California, USA
[4]Department of Atmospheric Sciences, Yonsei University, Seoul, Republic of Korea
[5]Royal Belgian Institute for Space Aeronomy (BIRA-IASB), Brussels, Belgium
[6]Institute of Environmental Physics, University of Bremen, Bremen, Germany
*now at: DLR - Institute for protection of maritime infrastructures, German Aerospace Center, Bremerhaven, Germany

*Correspondence to*: Rokjin J. Park (rjpark@snu.ac.kr)

**Abstract.** We describe a formaldehyde (HCHO) retrieval algorithm for the Geostationary Environment Monitoring Spectrometer (GEMS) that will be launched by the Korean Ministry of Environment in 2019. The algorithm comprises three steps: pre-processes, radiance fitting, and post-processes. The pre-processes include a wavelength calibration, and interpolation and convolution of absorption cross-sections; radiance fitting is conducted using a non-linear fitting method referred to as basic optical absorption spectroscopy (BOAS); and post-processes include air mass factor calculations and bias corrections for stripe patterns, background HCHO, and latitudinal biases. In this study, several sensitivity tests are conducted to examine the retrieval uncertainties using the GEMS HCHO algorithm. We evaluate the algorithm with the OMI Level 1B irradiance/radiance data by comparing our retrieved HCHO column densities with OMI HCHO products of the Smithsonian Astrophysical Observatory (OMHCHO) and the Belgian Institute for Space Aeronomy (OMI BIRA). Results show that OMI HCHO slant columns retrieved using the GEMS algorithm are in good agreement with OMHCHO, with correlation coefficients of 0.77–0.91 and regression slopes of 0.92–1.04 for March, June, September, and December 2005. Spatial distributions of HCHO slant columns from the GEMS



algorithm are consistent with the OMI BIRA products, but have relatively poorer correlation coefficients of 0.52 to 0.76 compared to those against the OMHCHO products. Also, we compare the satellite results with ground-based MAX-DOAS observations. OMI GEMS HCHO vertical columns are by 0.8–30% lower than those of MAX-DOAS at Haute-Provence

Observatory (OHP) in France, Bremen in Germany, and Xianghe in China. We find that the OMI GEMS retrievals have less bias than the OMHCHO and OMI BIRA products for OHP and Bremen in comparison with MAX-DOAS.

## 1 Introduction

Formaldehyde (HCHO) is mainly produced by the oxidation of non-methane volatile

organic compounds (NMVOCs), and it has been observed from space since the GOME instrument on ERS-2 satellite first began conducting column measurements in 1995 (Chance et al., 2000). The subsequent instrument, SCIAMACHY instrument on ENVISAT, collected continuous HCHO column data for 2002–2012 (Wittrock et al., 2006), and GOME-2A and -2B instruments have been conducting measurements from 2007 to the present day (De Smedt

et al., 2012; De Smedt et al., 2015). The OMI instrument was launched in 2004 and has provided global HCHO vertical column data with a higher spatial resolution of $13 \times 24$ km² at the nadir than that of former instruments. Furthermore, the TROPOMI equivalent to OMI has been offering consecutive data with an even finer spatial resolution of $7 \times 7$ km² at the nadir since 2017. There are thus more than 20 years of HCHO column data available from these

various instruments, which enable analyses to be conducted on the global changes of HCHO columns throughout this time period.

All of the low-orbiting sun-synchronous satellite measurements of HCHO columns mentioned above have played an important role in filling gaps for regions where limited (or no) in-situ measurements of HCHO have been made, and these measurements have been used to

constrain top-down estimates of biogenic and anthropogenic emissions of NMVOCs (Marais et al., 2012; Barkley et al., 2013; Stavrakou et al., 2014; Zhu et al., 2014). Together with satellite glyoxal measurements, HCHO satellite measurements have been used to distinguish dominant VOC sources (e.g., biogenic vs. anthropogenic) (DiGangi et al., 2012; Vrekoussis et al., 2010). In addition, the ratio of HCHO to nitrogen dioxide ($NO_2$) columns has been used to

determine NOx-limited or VOC-limited ozone production regimes (Martin et al., 2004; Duncan et al., 2010; Choi et al., 2012). Continuous HCHO column measurements from sun-





synchronous satellites are thus invaluable for evaluating and monitoring NMVOC emission trends over long-term periods.

However, as sun-synchronous satellites have observations periods of once or twice a day, they provide limited explorations of diurnal cycles and transboundary transports of air pollutants. Moreover, their coarse spatial resolutions make discerning local source emissions difficult. With the aim of overcoming the issues, Zhu et al. (2014) used the oversampling method and temporally averaged out pixels of OMI HCHO vertical columns over high-resolution grids of ~2 km, and Kim et al. (2016) developed a downscaling method for OMI $NO_2$ measurements by adopting the spatial distribution information from a regional air quality model. However, both these methods have inherent limitations: the former method involves a trade-off between spatial and temporal information and the latter method includes the uncertainties of emission distributions.

To tackle the limitations inherent in low-orbiting satellites measurements, environmental geostationary satellites will be launched in 2019 (or later) by South Korea and the United States, and in 2021 by European Union, to monitor air qualities over East Asia, North America, and Europe, respectively. Instruments on-board these geostationary satellites have superior spatial resolutions to those of sun-synchronous satellites and higher signal-to-noise ratios, and they will conduct column measurements of air pollutants every hour during the daytime. The Geostationary Environment Monitoring Spectrometer (GEMS) will be launched by South Korea, and it will measure radiances ranging from 300 to 500 nm every hour with a fine spatial resolution of $7 \times 8$ $km^2$ over Seoul in South Korea to monitor column concentrations of air pollutants including $O_3$, $NO_2$, $SO_2$, and HCHO, and aerosol optical properties. The measurements from GEMS will then be used in applications such as data assimilation of air quality forecasts and top-down constraints of air pollutant emissions.

This paper describes a GEMS retrieval algorithm for HCHO. It also presents an uncertainty analysis and an evaluation of the algorithm, which involves comparing OMI GEMS HCHO results with OMI HCHO products from the Smithsonian Astrophysical Observatory (OMHCHO) and those from the Belgian Institute for Space Aeronomy (OMI BIRA). In addition, OMI HCHO results are compared with those of MAX-DOAS ground observations. In Section 2, we describe the GEMS instrument and provide the theoretical basis for HCHO retrievals. In Section 3, sensitivity tests are conducted to examine the retrieval uncertainties; and in Section 4, we discuss an evaluation of HCHO results retrieved from the GEMS algorithm.



## 2 GEMS HCHO algorithm

### 2.1 GEMS instrument

GEMS is a scanning ultraviolet-visible spectrometer which will be launched by the Korean Ministry of Environment in 2019 on-board a geostationary satellite (GEO-KOMPSAT 2B), which also carries a Geostationary Ocean Color Imager-2 (GOCI-2). GEMS will be located at 128.2°E near the equator and will cover East and Southeast Asia (5°S-45°N, 75°-145°E). The instrument will conduct hourly measurements during the day (8 times) over the whole domain. It will scan one swath from south to north and then turn a scan mirror from east to west using an imaging time of 30 minutes and a transmission time of 30 minutes to enable GOCI-2 measurements for a 30 min period. Solar backscattered radiances will be measured in the 300–500 nm wavelength range with a spectral resolution of 0.6 nm and a wavelength interval of 0.2 nm. GEMS will provide finer spatial resolutions of $7 \times 8$ km$^2$ or less over Seoul, South Korea, compared to those of previous low-orbiting satellites (for example, OMI has a spatial resolution of $13 \times 24$ km$^2$ at nadir). The field of regards (FOR) and information about GEMS are shown in Fig. 1 and Table 1, respectively; detailed information about the GEMS instrument and algorithms for species other than HCHO are found elsewhere (Kim et al., 2018a).

### 2.2 HCHO algorithm description

Figure 2 is a flow chart of the HCHO retrieval algorithm for GEMS. The retrieval procedure consists of three steps: pre-processes, radiance fitting, and post-processes. The pre-processes begin with a wavelength calibration of Level 1B data (irradiance and radiance), and interpolation and convolution of absorption cross-sections at calibrated wavelength grid points. Radiance fitting is then conducted to derive the HCHO slant columns using a non-linear least square method. Finally, the post-processes include an air mass factor (AMF) calculation that employs a look-up table to convert HCHO slant columns to vertical columns, an assignment of data quality flags for each pixel, the removal of a possible stripe pattern along each cross-track position, and making corrections for background values and latitudinal biases. Each retrieval step is described in more detail in the following sections.





### 2.2.1 Wavelength calibration and GEMS bandpass function

Wavelength grid points of measured irradiances and radiances in a charge-coupled device (CCD) sensor are often shifted or squeezed, and such systematic biases due to wavelength shifts or squeezes need to be corrected when producing Level 1B data. However, as precise

wavelength alignments between irradiances/radiances and absorption cross-sections are required to achieve accurate radiance fitting, it is also necessary to conduct wavelength calibration during retrieval.

In wavelength calibration, the solar reference spectrum (Chance and Kurucz, 2010) is firstly convolved with a GEMS bandpass function and is then interpolated to the wavelength

grids of the measured spectrum. A convolved solar reference spectrum with wavelength shift parameters and polynomial parameters (Eq. 1) is then fitted to the measured irradiances and radiances in a broader fitting window (325.5–358.5 nm) than that of the radiance fitting for HCHO retrievals as follows,

$$I_R = (I_0^h * g)(\lambda + \Delta\lambda)P_{sc} + P_{bl}, \tag{1}$$

where $I_R$ is the modeled irradiance and radiance, $I_0^h$ is the solar reference spectrum with a high spectral resolution of 0.01 nm wavelength interval, $\Delta\lambda$ is the wavelength shift, $g(\lambda)$ is a bandpass function, and $P_{sc}$ and $P_{bl}$ are scaling and baseline polynomials, respectively. The

asterisk denotes the convolution procedure, as shown in Eq. 2.

$$(f * g)(\lambda) = \int_{-\infty}^{\infty} f(\Lambda)g(\lambda - \Lambda)d\Lambda \tag{2}$$

We use the GEMS bandpass functions for the convolution in wavelength calibration to

ensure consistency with the spectral resolutions of measured irradiances and radiances. GEMS bandpass functions are provided at seven center wavelengths ranging from -1.8 nm to 1.8 nm at center wavelengths, with wavelength sampling intervals of 0.06 nm (Fig. 3). The right panel in Fig. 3 shows bandpass functions averaged for spatial indices at 330 and 365 nm, and it also shows the relative differences between bandpass functions at 365 nm and 330 nm. The relative

differences are smaller near the wavelength center, but they increase over each wing of the function. For GEMS HCHO retrieval, we will conduct calibration for every spatial pixel of the sensor using the bandpass functions at 330 nm. However, as bandpass functions are not linear



with wavelength, it will be necessary to estimate uncertainties for the wavelength dependency of the bandpass functions after GEMS is launched.

### 2.2.2 Convolution and reference spectra sampling

Table 1 shows a summary of absorption cross-section datasets used in GEMS HCHO retrieval. In the retrieval algorithm, absorption cross-section data with fine spectral resolutions (for example, HCHO absorption cross-section data with a spectral resolution of 0.011 nm) are first convoluted with the bandpass functions described in Section 2.2.1, and they are then interpolated to the calibrated wavelength grids of measured radiances. Finally, radiance fitting accounts for attenuation of a reference spectrum (measured irradiance or radiance) by gas absorption using Eq. 3 with convoluted cross-section data as follows,

$$\text{attenuated radiance in radiance fitting} = (I_0^h * g)(\lambda)\exp\left(-(\tau^h * g)(\lambda)\right), \tag{3}$$

where the asterisks denote convolution (Eq. 2) and $\tau^h$ is the optical depth of interfering gases with fine spectral resolutions.

However, radiative transfer in the atmosphere occurs in a slightly different manner. Solar irradiance is firstly reduced by the absorption of interfering gases, and radiances are subsequently measured on discrete wavelength grids of an instrument with predetermined spectral resolutions, as shown in Eq. 4,

$$\text{attenuated radiance in reality} = ((I_0^h \exp(-\tau^h)) * g)(\lambda). \tag{4}$$

Therefore, the difference between measured and calculated radiances on discrete grids may provide biases in radiance fitting when sensors have limited spectral resolutions, which is referred to as the solar $I_0$ effect (Aliwell et al., 2002; Chan Miller et al., 2014).

We calculate pseudo absorption cross-sections to account for the differences in gas absorptions between reality and radiance fitting. We assume that the absorption process in radiance fitting is the same with that in reality, and the pseudo absorption cross-sections are computed using the following equation (Aliwell et al., 2002; Chan Miller et al., 2014),

$$\sigma_{ps}(\lambda) = \frac{1}{scd_{ref}}\ln\left(\frac{I_0^h(\lambda) * g(\lambda)}{[I_0^h(\lambda)\exp(-scd_{ref}\sigma^h(\lambda))] * g(\lambda)}\right), \tag{5}$$





where $\sigma_{ps}(\lambda)$ is a pseudo absorption cross-section, $scd_{ref}$ is a reference slant column density, and $\sigma^h(\lambda)$ is an absorption cross-section with a fine spectral resolution.

Although the corrected absorption cross-section can be applied to all the species, we only apply the correction to the $O_3$ absorption cross-section, which is the most important interfering species in the fitting window of HCHO retrievals, and we use an $O_3$ reference slant column density of 300 DU for the correction.

### 2.2.3 Radiance fitting

Three different methods have been used with sun-synchronous satellite measurements in previous HCHO retrievals: differential optical absorption spectroscopy (DOAS) method (De Smedt et al., 2008; De Smedt et al., 2012), a non-linearized fitting method, which is known as basic optical absorption spectroscopy (BOAS) (Chance et al., 2000; González Abad et al., 2015; González Abad et al., 2016), and principal component analysis (Li et al., 2015). Zhu et al. (2016) conducted an inter-comparison of HCHO vertical column densities retrieved using the three retrieval methods for four instruments such as OMI, GOME-2A, GOME-2B, and OMPS; they found that the different retrieval results were consistent in terms of temporal and spatial variations of HCHO columns in the southeastern United States.

To yield HCHO slant columns in this study, we use the BOAS method, which is based on a non-linearized form of the Lambert-Beer law, as shown in Eq. 6 (González Abad et al., 2015). One advantage of the BOAS method is that it uses unprocessed radiance data, and it is thus more intuitive than the widely used DOAS method, which uses a linearized form by taking the logarithm of radiance to irradiance and high-pass filtering the result. The modeled radiative equation is given as follows,

$$I(\lambda) = [(aI_0(\lambda) + c_r\sigma_r(\lambda)) \exp(- \textstyle\sum_i SCD_i\sigma_i) + c_{cm}\sigma_{cm}]P_{sc} + P_{bl}, \qquad (6)$$

where $a$ is an amplitude factor; $I_0$ is a reference spectrum (solar irradiance or radiance reference); $c_r\sigma_r(\lambda)$ is the contribution of the Ring effect; $\exp(-\sum_i SCD_i\sigma_i)$ is the contributions of all gas absorptions; $SCD_i$ and $\sigma_i$ are slant column densities and absorption cross-sections for species $i$, respectively; $c_{cm}\sigma_{cm}$ is the contribution of the common mode; and $P_{sc}$ and $P_{bl}$ are scaling and baseline polynomials, respectively, considering low



frequency variations due to Rayleigh and Mie scattering. Table 1 summarizes the detailed information used in the GEMS HCHO retrieval algorithm. Furthermore, the modeled spectrum is fitted to measured radiances using a non-linear least square method (Wedin and Lindström, 1987).

The common mode denotes fitting residuals caused by instrument properties which have not been determined from physical analysis. Accounting for the common mode can reduce fitting residuals and fitting uncertainties without affecting the retrieved slant columns (González Abad et al., 2015). The common mode for GEMS can be calculated by averaging fitting residuals at every cross-track over far east swaths, which are relatively clean regions.

**2.2.4 Air mass factor**

HCHO slant column densities ($\Omega_s$) from the radiance fitting are then converted to vertical columns ($\Omega_v$) with an AMF (Eq. 7), which is a correction factor for the light slant path to the vertical path. Previous studies have shown that AMF uncertainty is one of the crucial factors causing retrieval uncertainties (De Smedt et al., 2012 and 2018), and AMF uncertainties mainly

contribute to retrieval uncertainties over polluted areas because AMFs can be influenced by multiple factors (including HCHO vertical distribution, aerosol vertical distribution, and aerosol optical properties) (Chimot et al., 2016; Kwon et al., 2017; Hewson et al., 2015).

$$\Omega_v = \frac{\Omega_s}{AMF} \qquad (7)$$

An AMF can be decoupled with a scattering weight ($w_l$) and a vertical shape factor ($S_l$) of the target species (Eq. 8), which represent radiative sensitivity to the optical depth of the absorber and a partial column density profile normalized by total vertical column density, respectively, at each layer $l$ ($l = 1, 2, \ldots, n$) (Palmer et al., 2001). Scattering weights are

dependent on the solar zenith angle ($\theta_s$), viewing zenith angle ($\theta_v$), relative azimuth angle ($\theta_r$), surface albedo ($\alpha_s$), cloud top pressure ($p_{cld}$), and effective cloud fraction ($f_c$).

$$AMF = \sum_{l=1}^{n} w_l(lat, lon, month, \theta_s, \theta_v, \theta_r, \alpha_s, p_{cld}, f_c) S_l(lat, lon, month) \qquad (8)$$

The effective cloud fraction and cloud top pressure can be retrieved from GEMS with the assumption of a Lambertian cloud surface (cloud surface albedo = 0.8). In addition, the





radiative cloud fraction ($f_{rc}$) will be provided, and is defined by Eq. 9, where $I_{cld}$ and $I_{clr}$ are radiances over cloud and cloud-free surfaces, respectively.

$$f_{rc} = \frac{f_c I_{cld}}{(1-f_c)I_{clr}+f_c I_{cld}} \tag{9}$$

We also consider the presence of clouds in the AMF calculations: scattering weights in the partial cloudy scenes are linearly interpolated as a function of radiative cloud fractions with scattering weights for clear sky ($w_{l,nc}$) and fully covered cloudy sky ($w_{l,yc}$) (Eq. 10) (Martin et al., 2002; González Abad et al., 2015). The latter ($w_{l,yc}$) is calculated as a function of cloud

top pressures in the assumption of Lambertian cloud surface (see Table 1) with a cloud surface albedo of 0.8.

$$w_l = (1 - f_{rc})w_{l,nc}(lat, lon, month, \theta_s, \theta_v, \theta_r, \alpha_s) +$$
$$f_{rc}w_{l,yc}(lat, lon, month, \theta_s, \theta_v, \theta_r, p_{cld}) \tag{10}$$

For AMF calculations, we compile an AMF look-up table (LUT) at 340 nm as a function of the variables described in Eqs. 8–10 and Table 1. González Abad et al. (2015) showed that the wavelength dependence of scattering weights in the HCHO fitting window is small. Therefore, we ignore the wavelength dependence and use AMF values for one wavelength.

The AMF LUT is calculated using VLIDORT v2.6 (Spurr, 2006) with a priori data including temperature, pressure, and gas profiles ($O_3$, $NO_2$, $SO_2$, and HCHO), which are simulated from a 3-D chemical transport model (GEOS-Chem v9-01-02; Bey et al., 2001) driven by Modern-Era Retrospective Analysis for Research and Applications (MERRA) with 47 vertical levels and a 2° × 2.5° horizontal resolution, for 2014.

Figure 4 shows examples of scattering weights and vertical profile shapes from the AMF LUT in June with conditions of a solar zenith angle of 30°, a viewing zenith angle of 0°, a relative azimuth angle of 90°, a surface albedo of 0.1, a cloud top pressure of 800 hPa, and an effective cloud fraction of 0.3 over clean and polluted grids. Clean grids are classified as having a HCHO column density less than $3.0 \times 10^{15}$ molecules cm$^{-2}$ and a surface pressure higher than

990 hPa, and polluted grids have a HCHO column higher than $1.0 \times 10^{16}$ molecules cm$^{-2}$ and a surface pressure higher than 990 hPa. In Fig. 4, although scattering weights are not significantly changed, the normalized vertical profile (a vertical shape factor) over a polluted





area is larger near the surface compared to a clean area, which results in AMF values of 1.55 and 1.28 over the clean and polluted areas, respectively.

### 2.2.5 Post processes

Post processes include systematic bias corrections and a statistic data quality flag
calculation for each pixel. Systematic bias corrections include cross-track bias correction and background HCHO column correction. Cross-track biases appear as functions of each cross-track position when a solar irradiance is used as a reference spectrum and are attributed to cross-track variability of the measured irradiance. The biases for OMI are constant at different latitudes; therefore, the biases show as stripes in the along-track direction. The cross-track
biases are estimated by a polynomial fit through medians of HCHO slant columns for each cross-track position in clean sectors, which are the Pacific Ocean for OMI and easternmost swaths for GEMS. The biases are removed from all data measured on the same day for each cross-track position.

An alternative method that can be used to avoid the above-mentioned biases in the fitting
procedure is to use measured radiances over a clean background region (referred to as radiance references) as the reference spectrum in radiance fitting. As measured radiance includes instrument noise and attenuation by interfering gases in the background atmosphere, the interfering effects can be minimized in radiance fitting, which results in negligible cross-track biases.

In this case, the retrieved slant columns using a radiance reference are differential slant columns ($\Delta SCD = SCD - SCD_0$) and do not include background HCHO columns ($SCD_0$) that are mainly related to the oxidation of methane. To account for the background columns, we use HCHO vertical columns simulated in 2014 from a chemical transport model, GEOS-Chem (Bey et al., 2001) with a spatial resolution of $2° \times 2.5°$. For OMI products, simulated HCHO
vertical columns are zonally averaged over clean background regions for the reference sector (140-160°W, 90°S-90°N), except for Hawaii (154-160°W, 19-22°N), and are interpolated to 720 latitudinal grid points, with a resolution of 0.25° from 90°S to 90°N. For GEMS, we plan to use simulated HCHO columns over easternmost regions as GEMS reference sectors.

To derive slant columns from the model, AMF values over the reference sector ($AMF_0$)
are calculated with cloud information and geometric angles on the same orbit. Pixels for $AMF_0$ are selected with cloud fractions of less than 0.4, data quality flags of 0 (good pixels), and





retrieved slant columns of $\pm\ 1.0 \times 10^{16}$ molecules cm$^{-2}$. Corrected HCHO slant columns are formulated as differential slant columns and simulated slant columns in Eq. 11,

$$\Omega_s(i,j) = SCD_{corr}(i,j) = \Delta SCD(i,j) + AMF_0(lat)VCD_m(lat). \qquad (11)$$

However, we may also need to correct latitudinal biases resulting from O$_3$ interference dependent on latitudes. Previous studies conducted latitudinal bias corrections when using a radiance reference as the reference spectrum (González Abad et al., 2015; De Smedt et al., 2018). Latitudinal biases result from stratospheric O$_3$ distributions dependent on latitude; 
however, they can be corrected by subtracting slant columns retrieved for a radiance reference from the slant columns analyzed.

Figure 5 shows OMI HCHO slant columns from OMHCHO products (Fig. 5a) and the GEMS algorithm without and with latitudinal bias corrections (Fig. 5b and 5c). HCHO slant columns without latitudinal bias corrections (Fig. 5b) are retrieved larger in 5°N-25°N than 
OMHCHO products, but HCHO slant columns with the bias corrections are in better agreement with OMHCHO products. Figure 5d shows the absolute differences between OMI HCHO slant columns with and without latitudinal bias corrections from the GEMS algorithm. Slant columns with bias corrections increase at latitudes lower than 5°N and higher than 25°N but decrease at latitudes from 5°N-25°N. However, GEMS has south to north swaths, while sun-synchronous 
satellites have west to east swaths; therefore, latitudinal biases can be minimized when using a radiance reference for GEMS. However, the latitudinal biases need to be further investigated after GEMS is launched.

After correction of systematic biases and conversion to vertical column densities with AMFs, all pixels are flagged with vertical columns and fitting uncertainties. We assign a data 
quality flag of 0 for good pixels, where retrieved vertical columns plus two-times fitting uncertainties are positive. The pixels where retrieved vertical columns are negative within two-times fitting uncertainties, but positive within three-times fitting uncertainties, are assigned a data quality flag of 1, which represents suspected quality pixels. Pixels with negative vertical columns within three-times fitting uncertainties are designated bad quality pixels and given a 
data quality flag of 2, and missing values are flagged by -1. It is of note that these conditions are generous and that tighter conditions for good data may be required for analysis.





### 3 Uncertainty analysis

We use error and uncertainty terminology from the Guide to the Expression of Uncertainties in Measurements (GUM) (https://www.bipm.org/utils/common/documents/ jcgm/JCGM_100_2008_E.pdf). Following GUM, "error" means the difference between a

5   measurement result and a true value, and "uncertainty" means the dispersion of measurement values, such as a standard deviation and a full width at the half maximum. As there is a lack of exact values for HCHO vertical columns, we consider that the word "uncertainty" is more appropriate for use in our analysis.

Uncertainties in the retrieval steps mentioned in Section 2.2 are assumed to be

10   uncorrelated, because the steps are independently performed. Total uncertainty in HCHO vertical column density (VCD) using a radiance reference can be formulated as follows (Boersma et al., 2004; De Smedt et al., 2008),

$$\sigma_v^2 = \left(\frac{\partial \Omega_v}{\partial \Delta SCD} \sigma_s\right)^2 + \left(\frac{\partial \Omega_v}{\partial AMF} \sigma_{AMF}\right)^2 + \left(\frac{\partial \Omega_v}{\partial VCD_m} \sigma_m\right)^2 + \left(\frac{\partial \Omega_v}{\partial AMF_0} \sigma_{AMF_0}\right)^2, \quad (12)$$

where $\sigma$ is the uncertainty in each part, $\Omega_v$ is the vertical column density, and subscripts $v$, $s$, and $m$ represent vertical, slant, and model, respectively. The total uncertainty equation is transformed using Eqs. 7 and 11 into Eq. 13,

$$\sigma_v^2 = \frac{1}{AMF^2}\left[\sigma_s^2 + \frac{(\Delta SCD + AMF_0 VCD_m)^2}{AMF^2}\sigma_{AMF}^2 + AMF_0^2\sigma_m^2 + VCD_m^2\sigma_{AMF_0}^2\right]. \quad (13)$$

We analyze expected uncertainties for this GEMS algorithm by retrieving HCHO VCDs with OMI level 1B data. For the uncertainty analysis, we use absorption cross-section data and the fitting window summarized in Table 1, and use OMI slit function data (Dirksen et al., 2006).

25   However, when GEMS is launched, it will be necessary to conduct another uncertainty analysis for GEMS HCHO retrievals.





### 3.1 Uncertainties in slant columns

### 3.1.1 Random uncertainty

Uncertainties in slant columns result from random uncertainties ($\sigma_{rand}$) and systematic uncertainties ($\sigma_{sys}$) (Eq. 14) (De Smedt et al., 2018),

$$\sigma_s^2 = \sigma_{rand}^2 + \sigma_{sys}^2.$$ (14)

Random uncertainties are fitting uncertainties when yielding slant columns, and they mainly result from instrument noise. We can reduce random uncertainties by using measured radiances over clean areas as reference spectra instead of irradiances, as the use of measured radiances can minimize instrument noise and interference of $O_3$ and BrO in the stratosphere. In addition, averaging the resulting slant columns for individual pixels can reduce random uncertainties, but this is achieved at the expense of a loss of temporal and spatial resolution (De Smedt et al., 2018).

Random uncertainties can be calculated from root mean square (RMS) values of fitting residuals, degrees of freedom ($m - n$), and diagonal component of covariance matrix ($C_{j,j}$) for fitting parameters,

$$\sigma_{s,j}^2 = \frac{RMS^2}{m-n} C_{j,j} C_{j,j},$$ (15)

where $m$ and $n$ are the number of spectral grids and fitting parameters, respectively, and $j$ denotes specific species in fitting parameters.

To examine random uncertainties from the GEMS algorithm for HCHO, we select pixels with the following conditions: a main data quality flag of 0, an effective cloud fraction of less than 0.3, and a solar zenith angle of less than 60° in the GEMS field of view (5°S-45°N, 75°-145°E) for four months (March, June, September, and December) in 2005. Random uncertainties vary between $3.3 \times 10^{15}$ and $1.8 \times 10^{16}$ molecules cm$^{-2}$, corresponding to pixels with fitting RMS from $4.4 \times 10^{-4}$ to $2.0 \times 10^{-3}$, and accounting for 97% of all pixels. In comparison with OMHCHO retrievals, González Abad et al. (2015) showed fitting





uncertainties ranging from $6 \times 10^{15}$ molecules cm$^{-2}$ to 100% or larger of slant columns and RMS values from $0.4 \times 10^{-3}$ to $2.0 \times 10^{-3}$, which are similar to those from the GEMS algorithm.

### 3.1.2 Systematic uncertainty

Systematic uncertainties result from uncertainties of wavelength calibration, the bandpass
function for convolution, and absorption cross-sections. We estimate systematic uncertainties from sensitivity tests to parameters. First, systematic uncertainties associated with absorption cross-sections are estimated using alternative absorption cross-sections: we compare resulting slant columns to the baseline run with conditions in Table 1 for a one-month period (March 2005). In the analysis, we define an uncertainty as a standard deviation of differences between
the sensitivity run and baseline run. Absorption cross sections are convoluted and interpolated using the same spectral resolution and wavelength sampling to enable comparisons between datasets.

To test the retrieval sensitivity to HCHO absorption cross-sections, we use HCHO absorption cross-section datasets from Cantrell et al. (1990) instead of those of Chance and
Orphal (2011) which provide a rescaling of the datasets in Cantrell et al. (1990) to those of Meller and Moortgat (2000). Absorption cross-sections of Cantrell et al. (1990) are ~10% lower than those of Chance and Orphal (2011), and the differences are directly linked to slant column retrieval (Pinardi et al., 2013). Therefore, slant columns using Cantrell et al. (1990) are by a factor of 1.1 higher than those of the baseline run. The slant column changes are similar to
values from previous studies (Pinardi et al., 2013; De Smedt et al., 2018).

We conduct a sensitivity test to O$_3$ absorption cross-sections at 223 K and 293 K from Chehade et al. (2013). Uncertainties of these datasets at both temperatures are ~4% in the fitting window of the GEMS algorithm. Compared to the baseline run, use of the O$_3$ datasets of Chehade et al. (2013) at 223 K and 293 K changes the slant columns by ~20% and ~8% on
average, respectively, which provides uncertainties in slant columns of $1.4 \times 10^{15}$ and $0.57 \times 10^{15}$ molecules cm$^{-2}$. These uncertainties are larger than those of 13% and 5% from Pinardi et al. (2013) and De Smedt et al. (2018), respectively. It thus appears that the GEMS HCHO retrieval algorithm is the most sensitive to O$_3$ absorption, especially at low temperatures in the stratosphere, due to strong absorption in the ultraviolet.
A sensitivity test is conducted against the BrO dataset of Fleischmann et al. (2004), which is by ~9% lower than the baseline BrO dataset in the GEMS HCHO fitting window, and results



in ~4% slant column changes compared to the baseline run (with an uncertainty of $0.28 \times 10^{15}$ molecules cm$^{-2}$).

We then examine slant column uncertainties for $O_4$ and $NO_2$ absorption cross-sections. We use alternative $O_4$ absorption cross-sections from http://spectrolab.aeronomie.be/o2.htm that have differences of 28% compared to the dataset used in the baseline run. Due to the large uncertainties of the data compared to other absorption cross-section data, the resulting slant column changes are significant, ~24%, with an uncertainty of $1.6 \times 10^{15}$ molecules cm$^{-2}$. This uncertainty could thus be decreased by reducing the uncertainties of the $O_4$ datasets. Also, $O_4$ should be included in the large fitting interval for HCHO because it has strong peaks near 344.0 and 361 nm (De Smedt et al., 2015; Thalman and Volkamer, 2013). The $NO_2$ datasets from Burrows et al. (1998) are 2% larger than those in the baseline run; switching to them leads to ~5% slant column changes with uncertainty of $0.37 \times 10^{15}$ molecules cm$^{-2}$.

We also estimate the systematic uncertainties of slant columns for wavelength calibration and solar $I_0$ effects by using an alternative solar irradiance reference. As described in Section 2.2.1 and 2.2.2, a solar irradiance reference spectrum was used in the wavelength calibration and calculation of pseudo absorption cross-sections related with solar $I_0$ effect. An alternative reference spectrum from Kitt Peak National Observatory (Kurucz et al., 1984) is almost identical to that of the baseline run, but the resulting slant column changes are up to ~14% with uncertainty of $0.92 \times 10^{15}$ molecules cm$^{-2}$. We thus consider that the solar $I_0$ effect associated with the strongest interfering gas ($O_3$) is very sensitive to the reference spectrum.

The total systematic uncertainty of slant columns for the parameters discussed is 38% of the slant column densities on average for one month. This uncertainty is larger than that of De Smedt et al. (2018), which is 20% of the slant column densities. However, there are remaining slant column uncertainties resulting from uncertainties relating to other parameters, such as polynomial orders in Eq. 6, instrument bandpass functions, and temperature dependency of cross-sections. Therefore, we estimate the systematic uncertainty as being 38% of the slant columns, prior to conducting uncertainty analyses on other parameters.

### 3.2 Uncertainty in AMF

The AMF uncertainty can be estimated by each parameter in Eq. 16. We examine AMF uncertainties for surface albedo ($\alpha_s$), cloud top pressure ($p_c$), and effective cloud fraction ($f_c$) with a solar zenith angle of 30°, a viewing zenith angle of 30°, and a relative azimuth angle of





0°. Derivatives of AMF to parameters (slope) are calculated from AMF LUT, while the uncertainties of parameters ($\sigma_{\alpha_s}$, $\sigma_{p_c}$, and $\sigma_{f_c}$) are based on De Smedt et al. (2018),

$$\sigma_{AMF}^2 = \left(\frac{\partial AMF}{\partial \alpha_s}\sigma_{\alpha_s}\right)^2 + \left(\frac{\partial AMF}{\partial p_c}\sigma_{p_c}\right)^2 + \left(\frac{\partial AMF}{\partial f_c}\sigma_{f_c}\right)^2. \tag{16}$$

Figure 6a shows AMF sensitivities to surface albedos from AMF LUT with clear sky conditions ($f_c = 0$). AMF values increase linearly with an increase in surface albedo, and sensitivities of AMF to surface albedo are slightly higher for low surface albedos of less than 0.2. The HCHO mixing ratio is higher near the surface in polluted areas compared to clean

areas, which relates to lower AMF values in polluted areas with low surface albedos.

Clouds below or within HCHO layers increase photon path lengths due to multiple scattering, while clouds at high altitudes above HCHO layers shield photons from reaching the surface. Therefore, the AMF values decrease with a decrease in cloud top pressures (with an increase in cloud heights) due to fewer photons reaching near the surface (Fig. 6b). AMFs are

more sensitive to cloud top pressure near the surface; this implies AMF is more sensitive to the changes in photon path lengths with respect to multiple scattering than blocking by clouds near the surface.

Cloud roles are shown in Fig. 6c. For clouds at 800 hPa, the AMF values increase with an increase in the cloud fraction, which implies that more photons arrive at the near surface from

the multiple scattering by clouds with increasing cloud fractions. However, for clouds at high altitudes (500 hPa), AMF values decrease with an increase in cloud fractions due to the shielding effect of clouds.

Figure 7 shows AMF uncertainty as functions of each parameter. AMF uncertainties with respect to surface albedo are estimated using a surface albedo ($\sigma_{\alpha_s}$) uncertainty of 0.02

(Kleipool et al., 2008) and are uniform compared to those with respect to cloud properties. In cloud free conditions ($f_c = 0$ in Fig. 7c), uncertainties of surface albedo almost account for total AMF uncertainty. For cloudy conditions, we use the uncertainties of cloud top pressure ($\sigma_{p_c} = 50$ hPa) and cloud fraction ($\sigma_{f_c} = 0.05$) (Veefkind et al., 2016). In conditions where the surface albedo is less than 0.6 and the cloud top pressure is higher than 550 hPa, AMF

uncertainties for cloud top pressures account for most of the total AMF uncertainties (Fig. 7a and 7b). In the opposite conditions, the uncertainties of cloud fractions contribute to most of the total AMF uncertainties. In addition, AMF uncertainties for cloud top pressures account for



almost all the AMF uncertainties with a cloud fraction higher than 0.4 (Fig. 7c). Total AMF uncertainties result in 4% to 16% of HCHO vertical column uncertainties from Eqs. 13 and 16.

HCHO profile shapes, aerosol vertical distributions, and aerosol optical properties are also important sources of AMF uncertainty (Chimot et al., 2016; Kwon et al., 2017; Hewson et al., 2013). AMF uncertainties from HCHO profiles are estimated to range from 0.27–0.44; they are thus the largest uncertainties of all other parameters in Eq. 16. The AMF uncertainty in HCHO profiles leads to a 15%–63% uncertainty in HCHO vertical columns. However, we can minimize the a priori HCHO profile uncertainties by using averaging kernels.

Non-absorbing aerosols have similar roles to clouds in radiative transfer, and non-absorbing aerosols are implicitly considered when cloud information is used. However, absorbing aerosols, such as mineral dust and black carbon, counteract the effects of non-absorbing aerosols and clouds. In the GEMS domain, dust storms and biomass burning occur seasonally, and we may therefore need to consider using aerosol information and investigating the interaction between aerosols and clouds.

### 3.3 Uncertainty in background correction

The uncertainties of model results for background concentrations ($\sigma_m$) and AMFs over reference sectors ($\sigma_{AMF_0}$) contribute to uncertainties in background corrections ($\sigma_{bg}$) in Eq. 17, which represents the 3rd and 4th terms on the right-hand side of Eq. 13,

$$\sigma_{bg}^2 = \frac{1}{AMF^2}\left[AMF_0^2\sigma_m^2 + VCD_m^2\sigma_{AMF_0}^2\right]. \tag{17}$$

We estimate model uncertainties by the differences in model results between the GEMS algorithm and the OMHCHO products, and find that model HCHO vertical columns used in the GEMS algorithm are lower than those used in OMHCHO, with absolute differences ranging from $-1.3 \times 10^{15}$ to $6.1 \times 10^{14}$. In Section 4, we discuss the impacts of these differences on retrieved HCHO columns. Uncertainties related to AMF in the reference sectors are identical to those discussed in Section 3.2.





## 4 Results and Validation

In this section, our retrieval algorithm is validated by a comparison with products of other institutes. First, we yield HCHO column densities using the GEMS HCHO algorithm with OMI Level 1B data in the fitting options described in Table 1, and also include spectral undersampling in the fitting process (Chance et al., 2005). We conduct our calculations for one month of each season (March, June, September, and December) in 2005 to provide seasonal variation.

Figure 8 shows an example of retrieved HCHO optical depths and fitting residuals as functions of wavelengths for a pixel in Indonesia (March 23 2005; orbit 3655). The retrieved HCHO slant column is $3.2 \times 10^{16}$ molecules $cm^{-2}$, which is relatively high due to biomass burning in that region. Average slant column and random uncertainty for all pixels on the orbit are $7.6 \times 10^{15}$ and $6.9 \times 10^{15}$ molecules $cm^{-2}$, respectively, over the GEMS domain. The large random uncertainty results from random uncertainties of 100% or larger in pixels with low concentrations.

Figure 9 compares monthly mean slant columns retrieved using the GEMS algorithm and those of OMHCHO products (González Abad et al., 2015). We select pixels (1) with vertical columns ranging from $-5.0 \times 10^{15}$ to $10 \times 10^{16}$ molecules $cm^{-2}$, (2) a main data quality flag of 0, (3) an effective cloud fraction of less than 0.3, and (4) a solar zenith angle of less than 60. Latitudinal biases are also corrected using the method discussed in Section 2.2.5. Monthly mean slant column densities are weighted by uncertainties and overlapped areas between pixels and grid boxes with a horizontal resolution of $0.25° \times 0.25°$.

We find similar spatial patterns of HCHO slant columns in both products; this shows that high HCHO columns occur over Indonesia and the Indochina Peninsula in March and over Indonesia in September, owing to biomass burning and biogenic activities. In summer, HCHO enhancements over China are caused by biomass burning and the oxidation of biogenic and anthropogenic VOCs due to photochemical reactions. In addition, high HCHO slant column densities occur over the Pearl River Delta, where anthropogenic VOCs are emitted from petrochemical industries, cargo ports, paint production, and many other activities. The scatter-plot comparisons with OMHCHO products show that GEMS HCHO slant columns are in good agreement with OMHCHO products, with correlation coefficients of 0.77–0.91 and regression slopes of 0.92–1.04. The relative differences between GEMS slant columns and those of OMHCHO products are -3% to 0.1% on average over the domain.





However, some discrepancies are found despite overall good agreement between GEMS and OMHCHO products, and these are mainly related to the background correction from the different model results. Figure 10 shows the simulated HCHO vertical columns used for background corrections in OMHCHO (solid lines) and the GEMS algorithm (dashed lines),

respectively. Values show similar latitudinal variations with a peak in the tropics and gradual decreases in high latitudes, which reflects the photochemical production of HCHO; however the magnitudes slightly differ. The model results used in GEMS are smaller than those used in the OMHCHO products (especially within ± 20° latitudes) by -1.3 × $10^{15}$ molecules cm$^{-2}$ near 6°N in September. Both results were obtained from the same 3-D global chemical transport

model (GEOS-Chem) but different assimilated meteorological products were employed. HCHO vertical columns used in the OMHCHO products were from GEOS-Chem with GEOS-4 meteorological data (Millet et al., 2006), which have lower cloud optical depths near the equator compared to GEOS-3 and GEOS-5. Therefore, low cloud optical depths in the tropics resulted in faster methane oxidation and the production of more HCHO in relation to high

hydroxyl    radical    concentrations    ([http://wiki.seas.harvard.edu/geos-chem/index.php/GMAO_GEOS-4](http://wiki.seas.harvard.edu/geos-chem/index.php/GMAO_GEOS-4)). González Abad et al. (2016) also showed that the OMHCHO products are larger than OMI BIRA-IASB products near the equator, such as in Southeastern Asia, Tropical Africa, and the Amazon Basin.

We select four regions (Sumatra/Malaysia, the Indochina Peninsula, China, and Borneo),

where HCHO is abundant from biomass burning and biogenic and anthropogenic sources. Table 2 provides the relative differences between OMI GEMS HCHO slant columns and OMHCHO slant columns in these four regions. GEMS HCHO slant columns are 1% to 13% lower than those of OMHCHO in Sumatra/Malaysia and Borneo, because the differences in simulated HCHO column densities for background corrections are relatively large near the

equator compared to the mid-latitudes, as previously mentioned. In the Indochina Peninsula and China, however, the GEMS HCHO slant columns are 6% to 25% higher than the OMHCHO slant columns, although the simulated HCHO concentrations used in the GEMS algorithm for background corrections are lower than those used in OMHCHO.

We also compare OMI GEMS HCHO slant columns with OMI QA4ECV products from

BIRA-IASB (OMI BIRA) in Fig. 11 (De Smedt et al., 2018). The spatial distributions of the GEMS HCHO slant columns are consistent with OMI BIRA products and provide relatively poorer correlation coefficients of 0.52 to 0.76 compared to those of OMHCHO products. The



relative differences between GEMS and BIRA slant columns range from -11% to -22% on average over the GEMS domain.

This discrepancy could result from the radiance fitting. OMI BIRA products use the DOAS method while the GEMS algorithm use a non-linearized fitting method (BOAS) for radiance fitting. OMI BIRA uses a two-step method: radiance fitting is conducted in a broad fitting window (328.5–359.0 nm) to remove interferences from BrO and $O_4$, and HCHO slant columns are then derived in a narrow fitting window (328.5–346.0 nm) with fitted BrO slant columns. Also, different $O_3$ absorption cross sections (Serdyuchenko et al., 2014) are used in OMI BIRA at different temperatures (220 and 243 K), and a non-linear $O_3$ absorption effect (Puķīte et al., 2010) is included. Differences caused by $O_3$ absorption datasets is the largest term in the GEMS algorithm (as mentioned in Section 3.1.2); therefore, relatively lower slant columns in the GEMS algorithm compared to OMI BIRA may result from the different $O_3$ datasets employed and the non-linear effect.

The regional and seasonal magnitudes of relative differences between OMI GEMS and OMI BIRA slant columns are larger over the Indochina Peninsula and China and in summer and winter, respectively, compared to those over Sumatra/Malaysia and Borneo, and compared to in spring and fall (Table 2). Biomass burning often occurs in spring and fall within Sumatra/Malaysia and Borneo, and during the spring in the Indochina Peninsula, which results in strong HCHO enhancements (see Fig. 11). That implies that HCHO can be well-retrieved because of the abundant HCHO concentrations, regardless of the fitting method used.

We also compare satellite results with QA4ECV MAX-DOAS ground observations at Haute-Provence Observatory (OHP) in France, Bremen in Germany, and Xianghe in China (Table 3). MAX-DOAS data are daily averaged within the OMI overpass time (12:00–15:00 local time) and are collected at OHP in March, June, and September 2005, at Bremen in June and September 2005, and at Xianghe in May 2016, respectively. We collect OMI data pixels that are overlapped by a grid box of 0.25° at the center of the site location, and average values of OMI data are weighted by overlapped areas between pixels and grid boxes.

Comparisons of HCHO VCDs between MAX-DOAS and satellite products are shown in Fig. 12 and Table 3. MAX-DOAS HCHO VCDs averaged for each period are $7.9 \times 10^{15}$, $7.2 \times 10^{15}$, and $1.8 \times 10^{16}$ molecules cm$^{-2}$ at OHP, Bremen, and Xianghe, respectively. OMI GEMS results are 25%, 0.8%, and 30% lower than MAX-DOAS, respectively, at OHP, Bremen, and Xianghe. In particular, the GEMS algorithm yields lower HCHO VCDs by 45% and 27% in



June at OHP and Bremen, respectively, compared to those from MAX-DOAS. These lower values may be caused by the a priori HCHO profiles used in AMF calculation. In summer, HCHO is produced and concentrated near the surface, which results in lower AMFs (higher VCDs). Kim et al. (2018b) showed the anti-correlation between AMF values and the HCHO mixing ratios at 200 m above ground level.

OMHCHO products show similar tendencies as OMI GEMS, but they are much lower than those of OMI GEMS. OMI BIRA products are higher by 19%–66% than ground observations at all three sites. Of the three satellite products, the OMI GEMS products show the smallest relative differences at OHP and Bremen against MAX-DOAS, where HCHO VCDs are relatively low. However, compared to the other two sites, there are smaller differences for OMI BIRA products in Xianghe where there are high HCHO VCDs.

## 5 Conclusions and discussions

We have developed a GEMS HCHO algorithm based on a non-linearized fitting method and described the algorithm in detail. The GEMS HCHO algorithm consists of three steps: pre-processes, radiance fitting, and post-processes. Pre-processes include wavelength calibration, and interpolation and convolution of absorption cross-sections. In the radiance fitting, HCHO slant column densities are retrieved by minimizing the difference between calculated radiances with initial guesses of absorbing gases and measured radiances in HCHO fitting windows. Finally, AMF values are calculated from an AMF LUT, and bias corrections for stripe patterns, background concentrations, and latitudinal biases are conducted if necessary.

We estimated the uncertainties of slant columns, AMF, and background corrections. The random uncertainty of slant columns corresponds with OMHCHO products, but the systematic uncertainty is 38% of the slant columns, which is higher than that of De Smedt et al. (2018). However, the systematic uncertainty can be reduced by using up-to-date absorption cross-sections. AMF uncertainty with respect to surface albedo, cloud top pressure, and cloud fraction contributes to 4–16% of the HCHO vertical columns. AMF uncertainty with respect to HCHO profile height leads to 15%–63% uncertainty in the HCHO vertical columns, but this can be reduced by using averaging kernels.

OMI HCHO results from the GEMS algorithm were compared to OMHCHO products with consistent fitting conditions applied. OMI GEMS slant columns show good agreement with OMHCHO products, with correlation coefficients of 0.77–0.91 and regression slopes of





0.92–1.04. However, some differences were found despite good agreement between GEMS and OMHCHO products, and these are caused by the use of different model results for background corrections. Although both model results are simulated by GEOS-Chem, different assimilated meteorological products are employed. The simulated HCHO vertical columns used in OMHCHO products are from GEOS-Chem with GEOS-4 meteorological data, which have lower cloud optical depths near the equator compared to GEOS-3 and GEOS-5. Low cloud optical depths in the tropics result in faster methane oxidation and greater HCHO production caused by high hydroxyl radical concentrations. Therefore, OMHCHO slant columns are 1% to 13% higher than OMI GEMS slant columns in Sumatra/Malaysia and Borneo near the equator.

The spatial distributions of GEMS HCHO slant columns are consistent with OMI BIRA products, but they have relatively poorer correlation coefficients of 0.52 to 0.76 compared to the OMHCHO products. Relative differences between GEMS and BIRA slant columns range from -11% to -22% on average over the GEMS domain. We consider that the discrepancy between the GEMS and BIRA products may be caused by the different radiance fitting methods employed, the different $O_3$ absorption datasets, and consideration of the non-linear $O_3$ absorption effect.

We also compare satellite results with MAX-DOAS ground observations at OHP in France, Bremen in Germany, and Xianghe in China. HCHO VCDs from the GEMS algorithm are 25%, 0.8%, and 30% lower than MAX-DOAS, but the discrepancies at OHP and Bremen are the smallest among the three satellites against the in-situ data.

After GEMS is launched, several options need to be tested. As described in Section 2.2.1, as GEMS bandpass functions depend on wavelength, it will be necessary to estimate uncertainties resulting from wavelength dependence of bandpass functions. We may also need to conduct sensitivity tests to optimize the fitting window. We currently use a broad fitting window (328.5–356.0 nm). However, we may need to use a narrow fitting window to reduce interference from polarization effects because GEMS does not include a polarization scrambler (De Smedt et al., 2018). Also, latitudinal biases for a radiance reference can also be minimized in GEMS employing its south to north swaths, as we plan to use easternmost swaths (except for islands such as Japan and Indonesia) for a radiance reference. Therefore, we will further investigate latitudinal biases after GEMS is launched.



**Acknowledgements**

This subject is supported by Korea Ministry of Environment (MOE) as "Public Technology Program based on Environmental Policy (2017000160001)".

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

**Table 1. Summary of GEMS system attributes, parameters for radiance fitting, and parameters for the AMF look-up table.**

| GEMS system attributes | |
| --- | --- |
| Spectral range | 300–500 nm |





| | |
|---|---|
| Spectral resolution | 0.6 nm |
| Wavelength sampling | 0.2 nm |
| Spatial resolution | < 56 km$^2$ at Seoul |
| Observation coverage | ≥ 5000 × 5000 km$^2$ (5°S-45°N, 75°E-145°E) |
| Duty cycle | ≥ 8 times/day |
| Imaging time | ≤ 30 minutes |
| **Radiance fitting parameters** | |
| Fitting window (calibration window) | 328.5–356.5 nm (325.5–358.5 nm) |
| Solar reference spectrum | Chance and Kurucz (2010) |
| Absorption cross-sections | HCHO at 300 K (Chance and Orphal, 2011) |
| | O$_3$ at 228 K and 295 K (Malicet et al., 1995; Daumont et al., 1992) |
| | NO$_2$ at 220 K (Vandaele et al., 1998) |
| | BrO at 228 K (Wilmouth et al., 1999) |
| | O$_4$ at 294 K (Thalman and Volkamer, 2013) |
| Ring effect | Chance and Spurr (1997) |
| Common mode | On-line common mode from far east swaths |
| Scaling and baseline polynomials | 3$^{rd}$ order |
| **AMF look-up table parameters** | |
| Longitude (degree) (n=33) | 70 to 150 with 2.5 grid |
| Latitude (degree) (n=30) | -4 to 54 with 2.0 grid |
| Solar Zenith Angle (degree) (n=9) | 0, 10, 20, 30, 40, 50, 60, 70, 80 |
| Viewing Zenith Angle (degree) (n=9) | 0, 10, 20, 30, 40, 50, 60, 70, 80 |
| Relative Azimuth Angle (degree) (n=3) | 0, 90, 180 |
| Cloud Top Pressure (hPa) (n=7) | 900, 800, 700, 600, 500, 300, 100 |
| Surface Albedo (n=7) | 0, 0.1, 0.2, 0.3, 0.4, 0.6, 0.8, 1.0 |

**Table 2. Relative differences between OMI GEMS HCHO slant columns and OMHCHO and OMI BIRA slant columns in four regions.**

| Region | GEMS vs. OMHCHO | GEMS vs. OMI BIRA |
|---|---|---|





| | Mar. | Jun. | Sep. | Dec. | Mar. | Jun. | Sep. | Dec. |
|---|---|---|---|---|---|---|---|---|
| Sumatra/Malaysia (95°-110°E, 0°-7°N) | -7% | -12% | -8% | -4% | -0.5% | -18% | -6% | -15% |
| Indochina Peninsula (97°-110°E, 10°-20°N) | 8% | 11% | 9% | 25% | -7% | -20% | -20% | -17% |
| China (110°-120°E, 30°-40N) | 12% | 6% | 10% | 6% | -21% | -25% | -20% | -23% |
| Borneo (110°-118°E, 5°S-0°) | -7% | -13% | -7% | -1% | -9% | -13% | 0.4% | -18% |

**Table 3. Averaged HCHO VCDs (molecules cm⁻²) from MAX-DOAS ground observations and OMI satellite data at OHP in France, Bremen in Germany, and Xianghe in China. For satellites, mean**





values are weighted by overlapped area between satellite pixels and grid cells of 0.25° at locations.
Relative differences between OMI and MAX-DOAS are given in parentheses.

| Site* | Class | MAX-DOAS** | OMHCHO | OMI BIRA | OMI GEMS |
|---|---|---|---|---|---|
| OHP (44°N, 5.5°E) | Rural | $7.9 \times 10^{15}$ | $5.4 \times 10^{15}$ (-32%) | $1.3 \times 10^{16}$ (61%) | $5.9 \times 10^{15}$ (-25%) |
| Bremen (53°N, 9°E) | Urban | $7.2 \times 10^{15}$ | $6.0 \times 10^{15}$ (-17%) | $8.8 \times 10^{15}$ (21%) | $7.2 \times 10^{15}$ (-0.8%) |
| Xianghe (39°N, 117°E) | Sub-urban | $1.8 \times 10^{16}$ | $1.1 \times 10^{16}$ (-41%) | $2.2 \times 10^{16}$ (19%) | $1.3 \times 10^{16}$ (-30 %) |

* HCHO VCDs are averaged at OHP in March, June, September 2005; at Bremen in June and September 2005; and at Xianghe in May 2016.

5      **Fitting windows of 336–359 nm and 324–359 nm are used at OHP and Bremen, and at Xianghe, respectively.

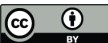



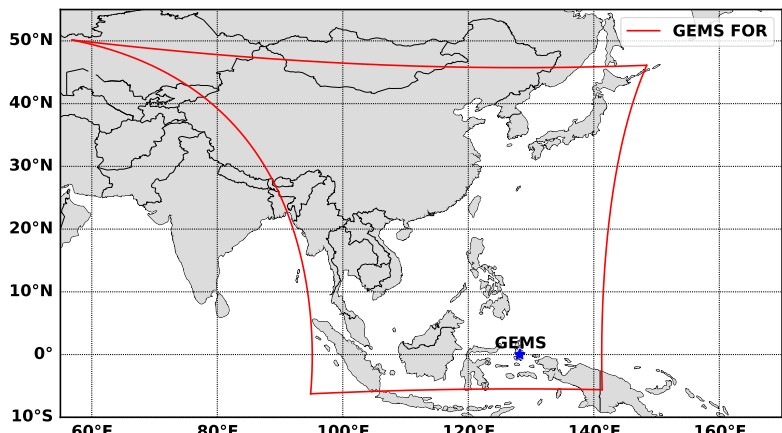

**Figure 1. GEMS field of regard (FOR) and GEMS location (blue star).**



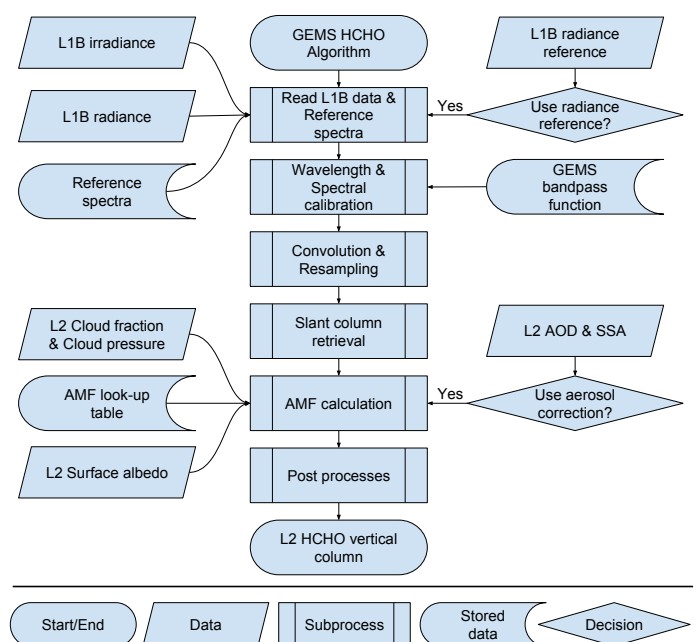

**Figure 2. Flow chart for GEMS HCHO algorithm.**



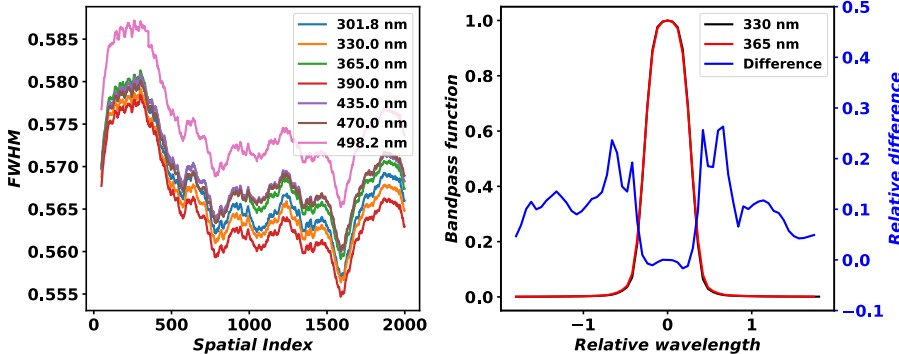

**Figure 3. Full width at half maximum (FWHM) of GEMS bandpass as a function of spatial index (left) and averaged bandpass functions to spatial index at 330 and 365 nm and relative differences (right).**





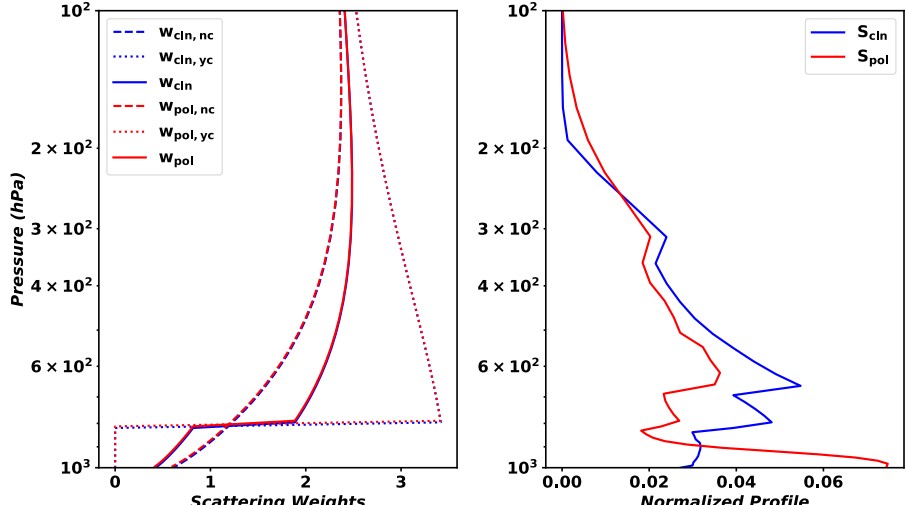

**Figure 4. Scattering weights (left) and normalized profiles (right) from the AMF LUT over clean (blue) and polluted (red) grids. Dashed and dotted lines in the left figure indicate scattering weights without and with cloud, respectively. The solid line in left figure indicates scattering weights calculated by Eq. 10.**

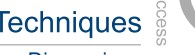

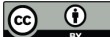

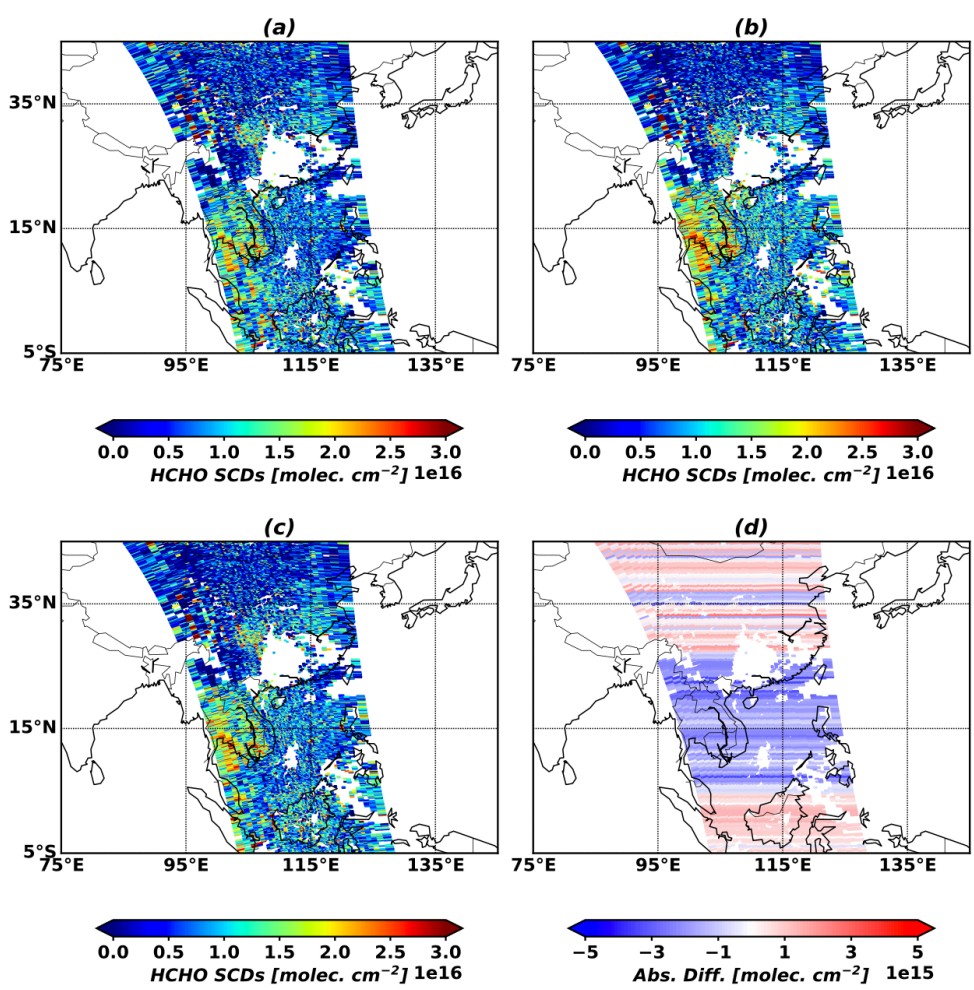

**Figure 5. HCHO slant column densities (March 8, 2005, orbit 3436): (a) OMHCHO products, (b) GEMS algorithm without latitudinal bias corrections and (c) GEMS with latitudinal bias corrections, and (d) differences ((c) – (b)).**





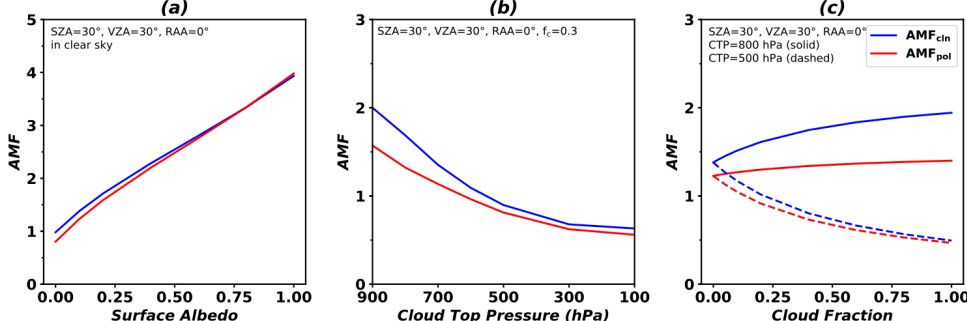

**Figure 6.** AMF variations as functions of (a) surface albedo, (b) cloud top pressure (CTP), and (c) effective cloud fraction ($f_c$) over clean (blue) and polluted (red) areas. Conditions of the AMF LUT are given in the figures. For sensitivity to surface albedo, cloud-free conditions are assumed. For sensitivity to cloud fraction, cloud top pressures are 800 hPa (solid line) and 500 hPa (dashed line).





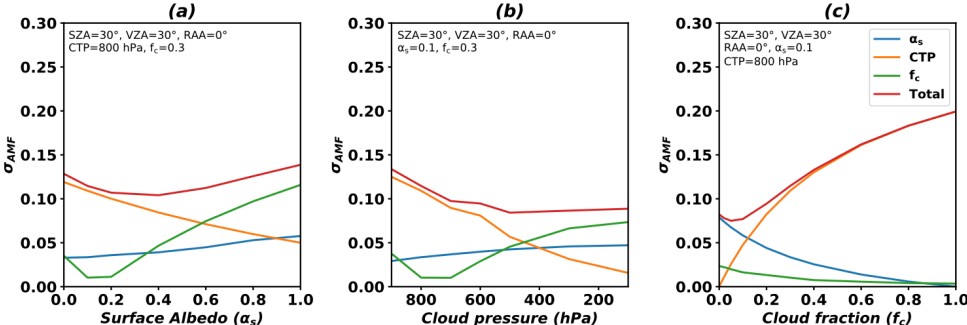

**Figure 7. AMF uncertainties from (a) surface albedo ($\alpha_s$; blue), (b) cloud top pressure (CTP; orange), and (c) effective cloud fraction ($f_c$; green) and total AMF uncertainty (red) to variations of each parameter. Conditions of the AMF LUT are given in the figures.**





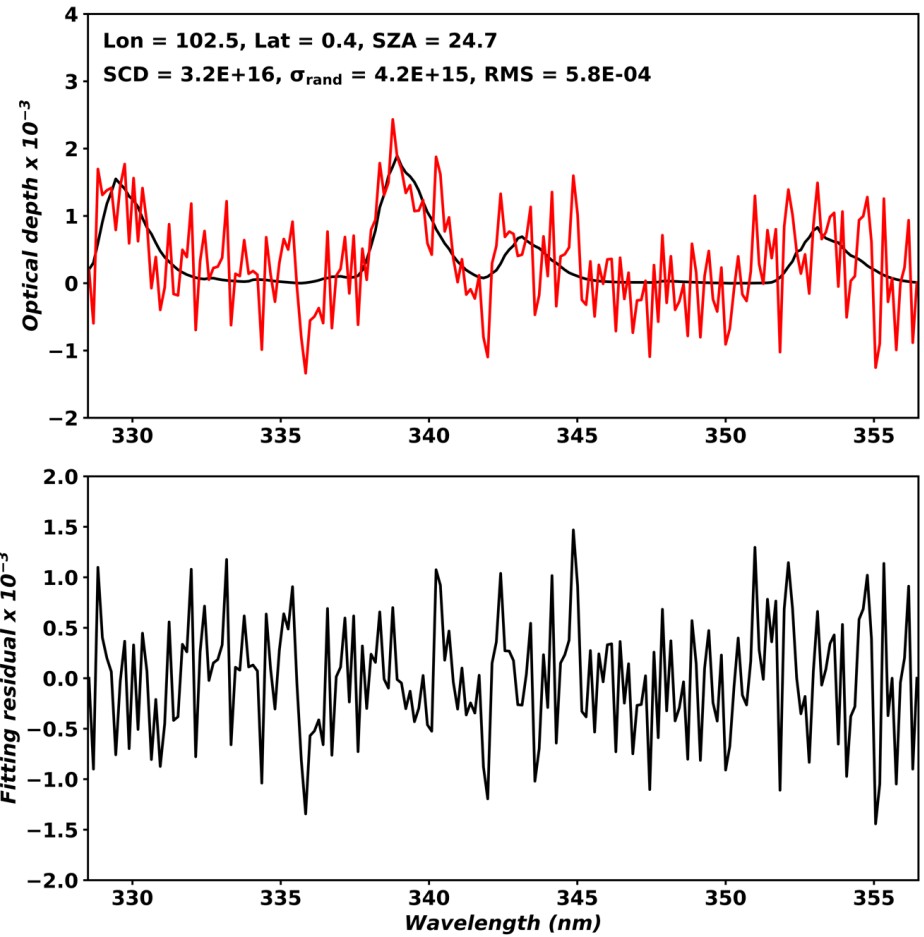

**Figure 8. Fitted HCHO optical depth (top) and fitting residuals (bottom) on a pixel (March 23, 2005; orbit 3655) with main data quality flag of 0 and effective cloud fraction less than 0.3. In the top panel, the black solid line indicates optical depth, and the red solid line indicates HCHO optical depth plus fitting residuals.**



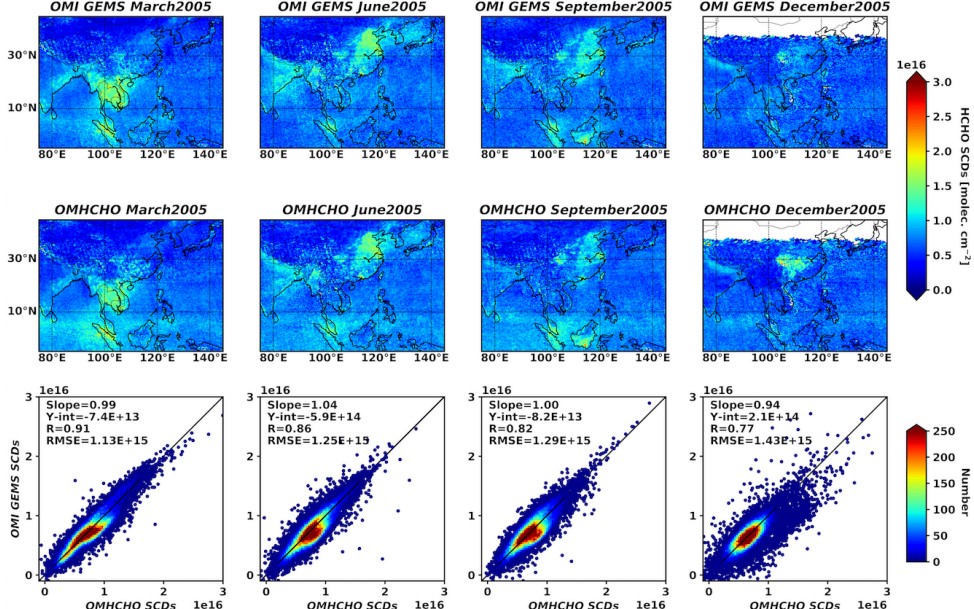

**Figure 9. Monthly mean slant columns from GEMS algorithm (1st row) and OMHCHO products (2nd row) for four months in 2005, and scatter plots between GEMS and OMHCHO. Statistics are given in the figures.**





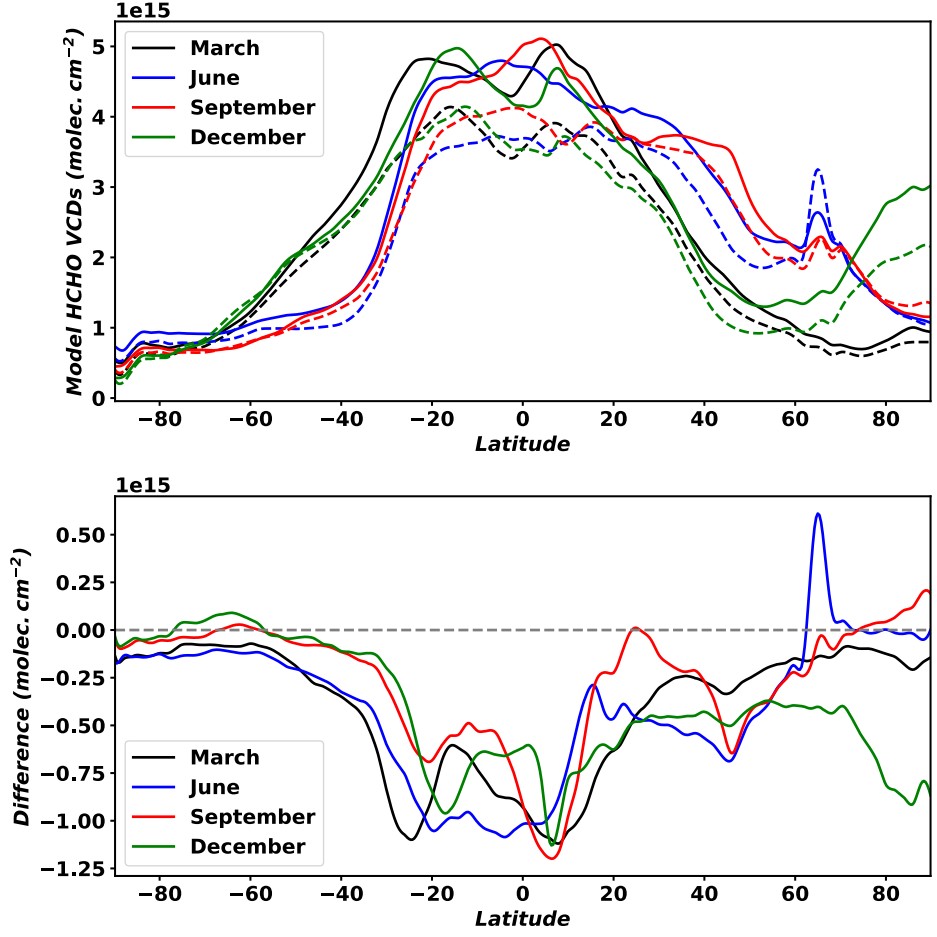

**Figure 10.** Simulated HCHO vertical column densities used in OMHCHO (solid lines) and GEMS algorithm (dashed lines) for background correction (top), and absolute differences of model results between GEMS algorithm and OMHCHO.





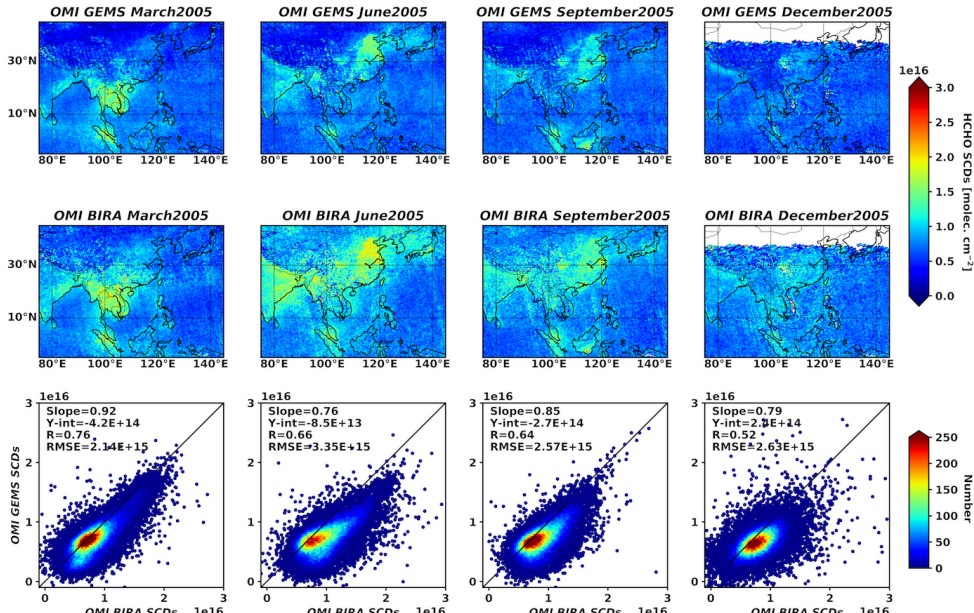

**Figure 11. Monthly mean slant columns from GEMS algorithm (1ˢᵗ row) and OMI BIRA products (2ⁿᵈ row) for four months in 2005, and scatter plots between GEMS and OMI BIRA. Statistics are given in the figures.**





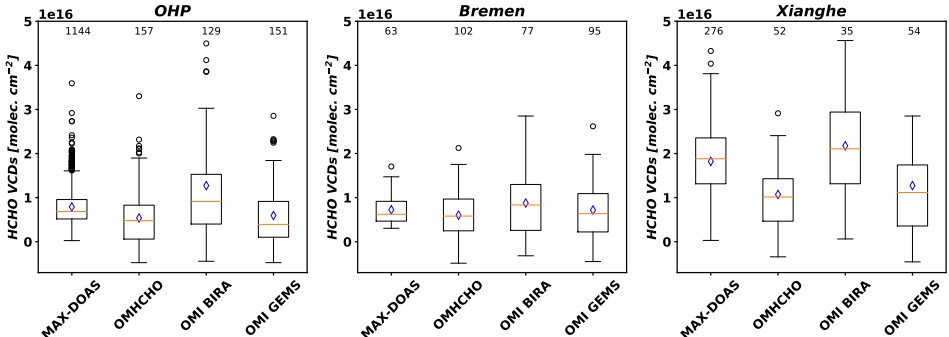

**Figure 12. HCHO vertical columns from MAX-DOAS, OMHCHO, OMI BIRA, and OMI GEMS at OHP (March, June, and September in 2005), Bremen (June, September in 2005), and Xianghe (May in 2016). Orange lines are median values for each dataset, and blue diamond markers are mean values. For satellites, mean values are weighted by overlapped area between satellite pixels and grid cells of 0.25° at locations. Boundaries of boxes indicate first and last quantiles of datasets.**