# Peer review of "Description of a formaldehyde retrieval algorithm for the Geostationary Environment Monitoring Spectrometer (GEMS)"

_Atmospheric Measurement Techniques, 2019_

## Referee Comment (RC1) · Anonymous Referee #1 · 13 Mar 2019

In this paper, Kwon et al. described the HCHO retrieval algorithm to be implemented with the Geostationary Environment Monitoring Spectrometer (GEMS). The authors discussed the three main steps in the algorithm (namely preprocessing, spectral fitting, and postprocessing), carried out uncertainty analysis, and also compared GEMS results (using OMI radiance data) with existing OMI HCHO results and MAX-DOAS at a few stations. Once launched, the HCHO data from GEMS can potentially be used in studies on regional air quality, and biomass burning in large areas over East and Southeast Asia. A paper providing detailed documentation of the retrieval algorithm is certainly of great interest to data users and the satellite remote sensing community. Overall, the paper is well organized, and figures and tables are mostly clear. I would

recommend publication in AMT after some clarifications (see below):

Specific comments: One would assume that there are some similarities and differences between the GEMS and OMHCHO algorithms. Some of these are discussed throughout the text, but it would be useful to have a table or a paragraph summarizing the different setups (and the resulting differences in HCHO) between the two instruments.

It appears that the background correction is a main contributor to the differences between GEMS and OMHCHO. And the two used different versions of GEOS-Chem for background VCDs. Is it possible to compare the model HCHO VCDs from the same model over the GEMS "background" area and OMHCHO "background" area? The easternmost part of GEMS FOR is still relatively close to Asia (and biomass burning and CH4 sources). A comparison may help to determine if the GEMS background is "background" enough.

Some symbols used in equations are not defined (immediately before or after the introduction of the equation), for example i and j in equation 11.

Page 7, Line 6: is 300 DU VCD of ozone for the pseudo cross section calculation?

Page 8, Line 7 and Figure 1: maybe you can define and plot the background areas?

Page 11, Line 15: would you expect that destriping would be necessary in the south/north direction?

It appears that latitudinal correction is implemented for the GEMS prototype algorithm – can the authors discuss uncertainties associated with this?

Figure 7: the figure is quite confusing – can the authors provide more detailed description and discussion?

Page 16, Line 29-34: can the authors briefly mention what kind of method/strategy/data will be used for aerosol correction, in the case of dust/smoke?

Figure 11 and related discussion: if ozone is an important contributor to the differences

between GEMS and BIRA OMI, maybe the authors can also compare the results from tropics and mid-latitude areas separately? One may expect somewhat better agreement between the two in the tropics? Or maybe the authors can run some test GEMS retrievals using the ozone cross section as used in BIRA retrievals?

———————————————

---

## Referee Comment (RC2) · Anonymous Referee #2 · 28 Mar 2019

This is a useful and timely manuscript on the algorithm for HCHO retrievals with the GEMS geostationary sensor that will observe the atmosphere in the near future over eastern Asia. It is useful because the algorithm is discussed in a step-by-step manner, and a thorough uncertainty assessment is included, and a comparison to independent data is provided. The discussion of the systematic component of the uncertainty is very strong. It is timely because the launch of GEMS is imminent, and the community would like to learn how retrievals are different or better than what we know from OMI and TROPOMI.

I recommend publication of the paper after the following issues are accounted or con-

sidered for.

Major issues

1. The paper focuses on testing the retrieval algorithm for OMI-type viewing conditions. It therefore remains unclear how the GEMS HCHO retrieval approach will account for diurnally varying measurement conditions. Surface reflectivity, HCHO profile shape, clouds will all change throughout the day, and it is unclear how these changes will affect the retrieval and their uncertainties. This is a major hiatus in this paper should be addressed.

2. Even though the instrument is still to be launched, the paper should give more information on the GEMS instrument and how its data will be explored. What is the anticipated signal-to-noise for the HCHO spectral window, or how would it compare to OMI and TROPOMI? How will the cloud retrieval from GEMS work? What surface reflectivity data will be used for the cloud and HCHO retrievals? How does the GEMS team address the issue of viewing geometry dependent surface reflectivity? These issues are not discussed, and thus the paper runs the risk of being read as just another OMI HCHO approach, i.e. of little specificity to GEMS.

3. The paper would be strengthened if the authors would provide a break up of the uncertainty budget for typical polluted and clean conditions, e.g. in the form of a table.

4. I'm missing a discussion of the GEOS-Chem 2x2.5 a priori profile shapes. These are much coarser than the 7x8 km2 viewing scenes, and this will result in a substantial AMF uncertainty. It is true that this can be accounted for via application of the averaging kernels, or by recomputing the AMFs with high-resolution profiles from a regional CTM or zoom-version of GEOS-Chem. In any case this issue should be discussed in more detail, and also included in the uncertainty budget.

5. It remains very much unclear how the latitude-bias is being determined. The text on page 11 (lines 9-11) is not clear, and the patterns shown in Figure 5(d) need explana-

tion.

Minor comments

P2, L12: suggest to remove 'instrument' after SCIAMACHY.

P3, L15: suggest to use air quality in the singular

P3, L16-17: compared to TROPOMI's 7x7 km2 pixels, the 7x8 km2 resolution from GEMS is not that superior, so I suggest to nuance that statement.

P6: eq. (3) and (4) – suggest to use that mathematical e rather than exp which reads as computer code.

Eq. (15) appears wrong.

Figure 7: what are the relative uncertainties in the AMF?

Validation: which spatio-temporal selection criteria were used?

---

## Referee Comment (RC3) · Anonymous Referee #3 · 4 Apr 2019

**Description of a formaldehyde retrieval algorithm for the Geostationary Environmental Monitoring Spectrometer (GEMS), Kwon et al., AMT, 2019**

**General Description:**

The authors describe the retrieval algorithm of formaldehyde (HCHO) for the future GEMS instrument and estimate the likely uncertainties and biases relative to OMI and ground-based MAX-DOAS measurements. The content is appropriate for AMT. Suggested changes, comments and concerns are included below.

**General Comments:**

It's not clear what's unique about the retrieval to GEMS. Seems more like a recapitulation of the OMI retrieval description paper of González Abad et al. (2015). A way to address this would be to assess the implication of the unique temporal component of GEMS (i.e. observations throughout the day) on uncertainties in the retrieval.

Throughout, use the standard symbol $\otimes$ for convolution. This will help clarify terms in equations that are confusing, as brackets are used to denote dependence, but also operators, e.g. $f \otimes g(\lambda)$ to replace $(f * g)(\lambda)$ in Equation (2) is clearer. Please correct these issues throughout.

Inconsistent use of wavelength dependence in equations. For example, why do $I_R$ and $I_0^h$ not depend on wavelength in Equations (1)-(4), but do in Equation (5)?

Many sub-sections in Section 2.2. are the same as in González Abad et al. (2015). Why not just refer the reader to that paper and only state aspects specific to GEMS and that are different between the two approaches?

It's not clear why Section 2.2.5 is relevant, as it describes bias corrections specific to OMI. Is it anticipated that the same bias corrections will be needed for GEMS? If this section is relevant, the readers could just be referred to González Abad et al. (2015) and this section be kept brief. It's also not clear why data quality flags are provided for a future product. This would only be important for the user when the data is ready for release.

Section 3 appears to just be testing uncertainties inherent in fitting parameters and retrieval terms that would be an issue for all space-based instruments measuring HCHO, rather than being specific to GEMS. Is there anything unique to GEMS (instrument configuration, viewing domain, repeat time etc.) that would increase or decrease sensitivity to these uncertainties relative to other instruments?

**Specific Comments:**

P2, Line 18: the spatial resolution of TROPOMI is finer than 7 x 7 km$^2$ for HCHO (De Smedt et al., 2018).

Equation (1): Why are $P_{sc}$ and $P_{bl}$ not dependent on wavelength?

P3, Line 22: Can aerosol optical properties be retrieved across this wavelength and for this type of instrument? Do the authors mean AOD and aerosol index (AI)?

Table 1: add references for these parameters as footnotes to point to consistency with existing retrievals.

P8, Lines 14-17: What about clouds (Millet et al., 2006)?

P18, Lines 8-14: Comment too on the implications of more observations over the same scene per day on uncertainty compared to OMI.

P18, Lines 26-28: Provide an appropriate reference for this statement.

Referencing: some references are missing the doi number (e.g., González Abad et al., 2015).

**References:**
González Abad et al., Atmos. Meas. Tech., 8, 19-32, 2015, doi:10.5194/amt-8-19-2015.
De Smedt et al., Atmos. Meas. Tech., 11, 2395–2426, 2018, doi:10.5194/amt-11-2395-2018.
Millet et al., J. Geophys. Res., doi:10.1029/2005JD006853, 2006

---

## Author Comment (AC1) · 15 May 2019

**Responses to Referee's Comments**

*We appreciate careful reading and lots of valuable comments.*

*We wrote referee's comments in black, our responses to comments in blue and italics, and the revised manuscript in red.*

**Referee #1:**

In this paper, Kwon et al. described the HCHO retrieval algorithm to be implemented with the Geostationary Environment Monitoring Spectrometer (GEMS). The authors discussed the three main steps in the algorithm (namely preprocessing, spectral fitting, and postprocessing), carried out uncertainty analysis, and also compared GEMS results (using OMI radiance data) with existing OMI HCHO results and MAX-DOAS at a few stations. Once launched, the HCHO data from GEMS can potentially be used in studies on regional air quality, and biomass burning in large areas over East and Southeast Asia. A paper providing detailed documentation of the retrieval algorithm is certainly of great interest to data users and the satellite remote sensing community. Overall, the paper is well organized, and figures and tables are mostly clear. I would recommend publication in AMT after some clarifications (see below):

Specific comments: One would assume that there are some similarities and differences between the GEMS and OMHCHO algorithms. Some of these are discussed throughout the text, but it would be useful to have a table or a paragraph summarizing the different setups (and the resulting differences in HCHO) between the two instruments.

*We used the same fitting options with OMHCHO products (González Abad et al., 2015), but auxiliary data such as model data for background corrections and AMF LUT are different. Also, we do not use undersampling correction and latitudinal bias correction for GEMS in default. For clarity, we only described GEMS fitting options in Section 2.2 and added Section 4.1 to explain differences of fitting options between GEMS and OMI as follows:*

**4.1 Retrieval of OMI HCHO**

GEMS fitting options described in Table 1 are largely consistent with those of OMHCHO products (González Abad et al., 2015). However, we do not include spectral undersampling (Chance et al., 2005) in the fitting process for GEMS, and reference sectors for a radiance reference are 143-150°E (shaded areas in Fig. 1). For OMI products, spectral undersampling needs to be included, and radiance references are from the Pacific Ocean as described in González Abad et al. (2015). We use simulated HCHO vertical columns for the background correction, which are zonally and monthly averaged over the reference sector (140-160°W, 90°S-90°N) except for Hawaii (154-160°W, 19-22°N).

In addition, we need to correct latitudinal biases for OMI. Previous studies explained that the latitudinal biases result from spectral interferences of BrO and $O_3$, whose concentrations are a function of latitude and are high in high latitudes (De Smedt et al., 2008; De Smedt et al., 2015; González Abad et al., 2015). Therefore, the latitudinal biases were corrected when a radiance reference was used as the reference spectrum (De Smedt et al., 2008; González Abad et al., 2015; De Smedt et al., 2018). We correct the latitudinal biases, which are slant columns retrieved for a radiance reference and are averaged as a function of latitude, by subtracting the biases from the corrected slant columns in Eq. 11. Figure 6 shows OMI HCHO slant columns from OMHCHO products (Fig. 6a) and the GEMS algorithm without and with latitudinal bias corrections (Fig. 6b and 6c). HCHO slant columns without latitudinal bias corrections (Fig. 6b) are retrieved larger in 5°N-25°N than OMHCHO products, but HCHO slant columns with the bias corrections are in better agreement with OMHCHO products. Figure 6d shows the absolute differences between OMI HCHO slant columns with and without latitudinal bias corrections from the GEMS algorithm as latitudinal biases. Slant columns with bias corrections increase at latitudes lower than 5°N and higher than 25°N but decrease at latitudes from 5°N-25°N. However, latitudinal biases can be minimized when using a radiance reference as a function of each cross-track position in the south to north direction for GEMS. In default fitting options, therefore, we do not include latitudinal correction and do not analyze uncertainty of latitudinal corrections in Section 3. However, a further investigation for the latitudinal biases needs to be required after GEMS is launched.

Figure 7 shows an example of retrieved HCHO optical depths and fitting residuals as functions of wavelengths for a pixel in Indonesia (March 23 2005; orbit 3655). The retrieved HCHO slant column is $3.2 \times 10^{16}$ molecules cm$^{-2}$, which is relatively high due to biomass burning in that region. Average slant column and random uncertainty for all pixels on the orbit are $7.6 \times 10^{15}$ and $6.9 \times 10^{15}$ molecules cm$^{-2}$, respectively, over the GEMS domain. The large random uncertainty of 100% or larger results from pixels with low concentrations, where averaged slant columns and random uncertainties are $2.2 \times 10^{15}$ and $6.2 \times 10^{15}$ molecules cm$^{-2}$.

It appears that the background correction is a main contributor to the differences between GEMS and OMHCHO. And the two used different versions of GEOS-Chem for background VCDs. Is it possible to compare the model HCHO VCDs from the same model over the GEMS "background" area and OMHCHO "background" area? The easternmost part of GEMS FOR is still relatively close to Asia (and biomass burning and CH4 sources). A comparison may help to determine if the GEMS background is "background" enough.

*In operation for GEMS, we will use easternmost regions as GEMS reference sectors (143-150℉), which are relatively clean areas for GEMS. Figure S1 shows model HCHO VCDs in GEMS background area (dashed) and OMI background area (solid) and absolute differences between the two.*
*We discussed it as follows:*

For GEMS, we plan to use simulated HCHO columns over easternmost regions (143-150°E) as GEMS reference sectors, which are shaded areas in Fig. 1. The GEMS reference sectors include part of islands near the equator and Japan but are relatively clean areas in south/north direction over the GEMS domain. In comparisons with background HCHO vertical columns over the Pacific Ocean for OMI (Fig. S1), annual mean of GEMS background columns over 4°S–45°N is $3.3 \times 10^{15}$ molecules cm$^{-2}$ slightly higher than that of OMI background columns ($3.2 \times 10^{15}$ molecules cm$^{-2}$), showing that we can use easternmost regions as background in the GEMS domain. Occasionally, local differences

between GEMS and OMI background columns can be as large as $3.8 \times 10^{15}$ molecules cm$^{-2}$ in the tropical region of the southern hemisphere due to biogenic activity and biomass burning, but the standard deviation of background values in that region is $5.1 \times 10^{14}$ molecules cm$^{-2}$ even lower than that of $1.2 \times 10^{15}$ molecules cm$^{-2}$ in the middle latitude (>30°N), indicating that the influences from biogenic activity and biomass burning can be corrected by model simulations.

[Figure]

10  **Figure S1. Simulated HCHO vertical column densities in GEMS background area (dashed lines) and OMI background area (solid lines) (top), and absolute differences between the two (bottom).**

15    Some symbols used in equations are not defined (immediately before or after the

introduction of the equation), for example i and j in equation 11.

*We explained symbols of i, j, and $VCD_m$ after Eq. 11 as follows:*

where i and j indicate pixel indices of cross and along tracks, respectively, and $VCD_m$ denotes a background vertical column density from the model.

Page 7, Line 6: is 300 DU VCD of ozone for the pseudo cross section calculation?

*The value of 300 DU is a slant column density of ozone for the pseudo cross section calculation and is $scd_{ref}$ in Eq. 5.*

Page 8, Line 7 and Figure 1: maybe you can define and plot the background areas?

*I defined longitudinal ranges (143-150°E) for common modes and radiance references. We updated Fig. 1:*

[Figure]

**Figure 1. GEMS field of regard (red), nominal daily scan (blue), full central scan (magenta), full western scan (cyan), and GEMS location (blue star). Shaded areas (143-150°E) are regions for radiance references and common mode.**

*We also updated Table 1 and modified some sentences related with reference sectors as follows:*

5 The common mode denotes fitting residuals caused by instrument properties which have not been determined from physical analysis. Accounting for the common mode can reduce fitting residuals and fitting uncertainties without affecting the retrieved slant columns (González Abad et al., 2015). The common mode for GEMS can be calculated by averaging fitting residuals at every cross-track over easternmost swaths (143-150°E)

10 shown as shaded areas in Fig. 1, which are relatively clean regions.

Table 1 summarizes the detailed information used in the GEMS HCHO retrieval algorithm. We follow fitting options in González Abad et al. (2015). We use measured radiances as the reference spectrum, called a radiance reference, and measured radiances are averaged over the easternmost swaths (143-150°E; shaded areas in Fig. 1) for a day

15 as a function of cross-track positions in the south to north direction. Background corrections are required when we use a radiance reference and are discussed in Section 2.2.5. Also, GEMS has cross-track swaths in the south to north directions while instruments such as OMI and TROPOMI have west to east swath. Therefore, latitudinal biases resulting from BrO and $O_3$ latitude-dependent interferences can be minimized for

20 GEMS and are discussed in Section 4.1.

**Table 1. Summary of GEMS system attributes, parameters for radiance fitting, and parameters for the AMF look-up table.**

| GEMS system attributes | |
| --- | --- |
| Spectral range | 300–500 nm |
| Spectral resolution | < 0.6 nm |
| Wavelength sampling | < 0.2 nm |
| Signal-to-noise ratio | > 720 at 320 nm |
| | > 1500 at 430 nm |
| Field of regard | $\geq 5000$ (N/S) $\times 5000$ (E/W) $km^2$ |
| | (5°S-45°N, 75°E-145°E) |

| | |
|---|---|
| Spatial resolution (at Seoul) | < 3.5 × 8 km$^2$ for aerosol |
| | < 7 × 8 km$^2$ for gas |
| Duty cycle | ~ 8 times/day |
| Imaging time | ≤ 30 minutes |
| **Radiance fitting parameters[a]** | |
| Fitting window (calibration window) | 328.5–356.5 nm (325.5–358.5 nm) |
| Radiance reference | Measured radiances from far east swaths (143-150°E) for a day |
| Solar reference spectrum | Chance and Kurucz (2010)[b] |
| Absorption cross-sections | HCHO at 300 K (Chance and Orphal, 2011) |
| | O$_3$ at 228 K and 295 K (Malicet et al., 1995; Daumont et al., 1992) |
| | NO$_2$ at 220 K (Vandaele et al., 1998)[b] |
| | BrO at 228 K (Wilmouth et al., 1999) |
| | O$_4$ at 293 K (Thalman and Volkamer, 2013)[b] |
| Ring effect | Chance and Spurr (1997)[b] |
| Common mode | On-line common mode from easternmost swaths (143-150°E) for a day |
| Scaling and baseline polynomials | 3$^{rd}$ order |
| **AMF look-up table parameters** | |
| Longitude (degree) (n=33) | 70 to 150 with 2.5 grid |
| Latitude (degree) (n=30) | -4 to 54 with 2.0 grid |
| Solar Zenith Angle (degree) (n=9) | 0, 10, 20, 30, 40, 50, 60, 70, 80 |
| Viewing Zenith Angle (degree) (n=9) | 0, 10, 20, 30, 40, 50, 60, 70, 80 |
| Relative Azimuth Angle (degree) (n=3) | 0, 90, 180 |
| Cloud Top Pressure (hPa) (n=7) | 900, 800, 700, 600, 500, 300, 100 |
| Surface Albedo (n=7) | 0, 0.1, 0.2, 0.3, 0.4, 0.6, 0.8, 1.0 |

[a] GEMS fitting parameters follow González Abad et al. (2015). However, undersampling is not included in the fitting parameters for GEMS, and reference sectors for radiance reference and common mode are different.

[b] The datasets are used in QA4ECV retrievals. Please refer to De Smedt et al. (2018) for other datasets and fitting options.

Page 11, Line 15: would you expect that destriping would be necessary in the south/north direction?

*GEMS has cross tracks (swaths) in the south to north directions. When we use radiance references, we make radiance references as a function of cross-track positions, already including effects of latitudinal dependent ozone absorption. Therefore, we think stripe patterns or biases would not be expected in the south/north direction.*

*We explained latitudinal biases in more detail as follows:*

In addition, we need to correct latitudinal biases for OMI. Previous studies explained that the latitudinal biases result from spectral interferences of BrO and $O_3$, whose concentrations are a function of latitude and are high in high latitudes (De Smedt et al., 2008; De Smedt et al., 2015; González Abad et al., 2015). Therefore, the latitudinal biases were corrected when a radiance reference was used as the reference spectrum (De Smedt et al., 2008; González Abad et al., 2015; De Smedt et al., 2018). We correct the latitudinal biases, which are slant columns retrieved for a radiance reference and are averaged as a function of latitude, by subtracting the biases from the corrected slant columns in Eq. 11.

…

However, latitudinal biases can be minimized when using a radiance reference as a function of each cross-track position in the south to north direction for GEMS. In default fitting options, therefore, we do not include latitudinal correction and do not analyze uncertainty of latitudinal corrections in Section 3. However, a further investigation for the latitudinal biases needs to be required after GEMS is launched.

It appears that latitudinal correction is implemented for the GEMS prototype algorithm – can the authors discuss uncertainties associated with this?

*As we mentioned above, latitudinal corrections are not included for GEMS. For clarity, we remained explanation only related with GEMS in Section 2.2 and explained fitting options for OMI in Section 4.1. Please refer to the first answer.*

Figure 7: the figure is quite confusing – can the authors provide more detailed description and discussion?

*AMF uncertainties are as functions of parameters in the AMF LUT and are sensitive to measurement conditions. Therefore, figure 7 is too confusing to explain the contributions of parameters so that we deleted Fig. 7. However, we added Table 2 to describe retrieval uncertainties of GEMS HCHO VCDs due to AMF uncertainties. We discussed it as follows:*

Table 2 summarizes estimated retrieval uncertainties of GEMS HCHO VCDs due to AMF uncertainties as functions of surface albedos, cloud top pressures, and cloud fractions. Values are calculated assuming conditions with solar zenith angle of 30°, viewing zenith angle of 30°, relative azimuth angle of 0°, cloud fractions less than 0.3, and a profile height of 700 hPa. Uncertainties of HCHO VCDs due to AMF uncertainties can be as large as 20% and 24% of HCHO VCDs in clean and polluted areas, respectively. Maximum values occur for conditions with low surface albedo and clouds at high altitudes, and high cloud fractions, but they do not differ much between clean and polluted areas. However, AMF driven HCHO uncertainty with respect to the profile height in polluted areas is higher than that in clean areas, implying that accurate HCHO profile information in polluted areas is important for the GEMS HCHO retrieval. We can minimize the a priori HCHO profile uncertainties by using averaging kernels.

**Table 2. Retrieval uncertainties of GEMS HCHO VCD due to AMF uncertainties as functions of surface albedos, cloud top pressures, cloud fractions, and HCHO profile heights for clean and polluted areas. Values are calculated for conditions with solar zenith angle of 30°, viewing zenith angle of 30°, relative azimuth angle of 0°, cloud fractions less than 0.3, and a profile height of 700 hPa.**

| AMF contribution to HCHO VCD uncertainty | Clean | Polluted |
|---|---|---|
| Surface albedo ($\alpha_s$) | 1-10% | 1-12% |
| Cloud top pressure ($p_c$) | 0-11% | 0-11% |
| Cloud fraction ($f_c$) | 0-19% | 0-17% |

| | | |
|---|---|---|
| HCHO height ($p_h$) | 0-11% | 0-17% |
| Total | 2-20% | 3-24% |

Page 16, Line 29-34: can the authors briefly mention what kind of method/strategy/data will be used for aerosol correction, in the case of dust/smoke?

*We briefly referred to methods for aerosol correction. We added sentences below the paragraph as follows:*

We plan to update our AMF LUT as a function of aerosol optical depth, single scattering albedo, and aerosol height, which will be retrieved in GEMS, to account for the effect of absorbing aerosols. On-line AMF calculation can also be used for aerosol correction with cloud information and model simulation (Lin et al., 2014).

Figure 11 and related discussion: if ozone is an important contributor to the differences between GEMS and BIRA OMI, maybe the authors can also compare the results from tropics and mid-latitude areas separately? One may expect somewhat better agreement between the two in the tropics? Or maybe the authors can run some test GEMS retrievals using the ozone cross section as used in BIRA retrievals?

*We found that differences between GEMS and QA4ECV (BIRA) OMI results are caused by polynomial orders. We tested effects of polynomial orders on slant columns. Also, we tested effects of different $O_3$ absorption cross-sections and non-linear $O_3$ absorption.*
*We found that using the $4^{th}$ polynomial order improves both correlation coefficient and regression slopes although OMI GEMS HCHO slant columns are higher than those in QA4ECV (Fig. S2). Using different $O_3$ datasets and considering non-linear effects of $O_3$ improve statistics and relative differences between GEMS and QA4ECV products (Table S1 and S2).*
*We discussed them in the manuscript as follows:*

The discrepancy between the two products could result from the radiance fitting. The OMI QA4ECV products use the DOAS method while the GEMS algorithm uses a non-linearized fitting method (BOAS) for radiance fitting. We also find that polynomial orders accounting for Rayleigh and Mie scatterings are important factors, causing differences between the two products. Retrieved slant columns using the 4th polynomial order are in better agreement with the QA4ECV products (Fig. S2). Both correlation coefficient and regression slope are improved although OMI GEMS HCHO values are higher than those of the QA4ECV. We use the 4th order polynomial instead of the 5th order used in the QA4ECV products because slant columns retrieved using the 5th order in the GEMS algorithms are much higher than the QA4ECV products.

Also, different $O_3$ absorption cross sections (Serdyuchenko et al., 2014) are used in the OMI QA4ECV at different temperatures (220 and 243 K), and a non-linear $O_3$ absorption effect (Puķīte et al., 2010) is included in the OMI QA4ECV. We examine the $O_3$ effects on retrieved slant columns in GEMS algorithm using $O_3$ datasets used in QA4ECV and considering a non-linear $O_3$ absorption effect. Correlation coefficient and regression slopes are slightly improved (Table S1), and relative differences in the four regions defined above are slightly reduced in most seasons and regions (Table S2).

**Table S1. Spatial correlation coefficients and slopes between OMI GEMS and OMI QA4ECV. Left values are statistics in Fig. 10, and right values are statistics of OMI GEMS using $O_3$ datasets used in QA4ECV and considering non-linear $O_3$ absorption effects.**

| Statistics | OMI GEMS vs. OMI QA4ECV | | | |
|:---:|:---:|:---:|:---:|:---:|
| | Mar. | Jun. | Sep. | Dec. |
| R | 0.76 / 0.75 | 0.66 / 0.70 | 0.64 / 0.67 | 0.52 / 0.54 |
| Slope | 0.92 / 1.02 | 0.76 / 0.82 | 0.85 / 0.91 | 0.79 / 0.84 |

**Table S2. Relative differences between OMI GEMS HCHO slant columns and OMI QA4ECV**

slant columns in four regions. Left values are relative differences in Table 3 and right values are relative differences of OMI GEMS using O$_3$ datasets used in QA4ECV and considering non-linear O$_3$ absorption effects.

| Region | OMIGEMS vs. OMI QA4ECV | | | |
|---|---|---|---|---|
| | Mar. | Jun. | Sep. | Dec. |
| Sumatra/Malaysia (95°-110°E, 0°-7°N) | -0.5% / 3% | -18% / -17% | -6% / -4% | -15% / -13% |
| Indochina Peninsula (97°-110°E, 10°-20°N) | -7% / -3% | -20% / -18% | -20% / -15% | -17% / -12% |
| China (110°-120°E, 30°-40N) | -21% / -25 % | -25% / -20% | -20% / -14% | -23% / -23% |
| Borneo (110°-118°E, 5°S-0°) | -9% / -5% | -13% / -9% | -0.4% / 5% | -18% / -16% |

[Figure]

**Figure S2. The same as Fig.10 but OMI GEMS products are retrieved using the 4$^{th}$ order polynomial instead of the 3$^{rd}$ order polynomial in default fitting options.**

---

## Author Comment (AC2) · 15 May 2019

**Responses to Referee's Comments**

*We appreciate careful reading and lots of valuable comments.*

*We wrote referee's comments in black, our responses to comments in blue and italics,*

5  *and the revised manuscript in red.*

**Referee #2:**

This is a useful and timely manuscript on the algorithm for HCHO retrievals with the GEMS geostationary sensor that will observe the atmosphere in the near future over

10  eastern Asia. It is useful because the algorithm is discussed in a step-by-step manner, and a thorough uncertainty assessment is included, and a comparison to independent data is provided. The discussion of the systematic component of the uncertainty is very strong. It is timely because the launch of GEMS is imminent, and the community would like to learn how retrievals are different or better than what we know from OMI and TROPOMI.

15  I recommend publication of the paper after the following issues are accounted or considered for.

Major issues

1. The paper focuses on testing the retrieval algorithm for OMI-type viewing conditions.

20  It therefore remains unclear how the GEMS HCHO retrieval approach will account for diurnally varying measurement conditions. Surface reflectivity, HCHO profile shape, clouds will all change throughout the day, and it is unclear how these changes will affect the retrieval and their uncertainties. This is a major hiatus in this paper should be addressed.

25  *Thanks for your suggestions about our weakness. As you mentioned, HCHO products can be sensitive to diurnally varying parameters such as surface reflectivity, HCHO profile shape, and clouds. Even though these parameters can affect the radiance fitting, we can estimate effects of variations of the parameters on AMF. Uncertainty of AMF as a function of the parameters was discussed in Section 3.2.*

30  *To examine sensitivity of AMF to HCHO profile height, we added Fig. 5d.*

[revised manuscript text omitted]

2. Even though the instrument is still to be launched, the paper should give more information on the GEMS instrument and how its data will be explored. What is the anticipated signal-to-noise for the HCHO spectral window, or how would it compare to OMI and TROPOMI?

*Requirements of the signal-to-noise ratio for GEMS are greater than 720 at 320 nm and 1500 at 430 nm for natural spatial resolutions of 3.5 × 8 km². Required signal-to-noise-ratios of OMI are 1450 in 335-365 nm, 700 in 365-420 nm, and 2600 in 420-450 nm for spatial resolution of 13 × 24 km² (OMI L1B ATBD). Signal-to-noise ratios of TROPOMI are 800-1000 in 310-405 nm and 405-500 nm (Veefkind et al., 2012). GEMS signal-to-noise ratios are comparable with those of OMI and TROPOMI. We added sentences and updated Table 1 as follows:*

Requirements of signal-to-noise ratio for GEMS are 720 and 1500 at 320 and 430 nm, respectively, for natural spatial resolutions (3.5 × 8 km² over Seoul). However, pixels are co-added in order to increase signal-to-noise ratio, and GEMS will provide spatial resolutions of 7 × 8 km² or less over Seoul, South Korea for trace gases.

**Table 1. Summary of GEMS system attributes, parameters for radiance fitting, and parameters for the AMF look-up table.**

| GEMS system attributes | |
| --- | --- |
| Spectral range | 300–500 nm |
| Spectral resolution | < 0.6 nm |
| Wavelength sampling | < 0.2 nm |
| Signal-to-noise ratio | > 720 at 320 nm |
| | > 1500 at 430 nm |
| Field of regard | ≥ 5000 (N/S) × 5000 (E/W) km² |
| | (5°S-45°N, 75°E-145°E) |
| Spatial resolution (at Seoul) | < 3.5 × 8 km² for aerosol |
| | < 7 × 8 km² for gas |
| Duty cycle | ~ 8 times/day |
| Imaging time | ≤ 30 minutes |
| **Radiance fitting parameters[a]** | |
| Fitting window (calibration window) | 328.5–356.5 nm (325.5–358.5 nm) |
| Radiance reference | Measured radiances from far east swaths (143-150°E) for a day |

| | |
|---|---|
| Solar reference spectrum | Chance and Kurucz (2010)[b] |
| Absorption cross-sections | HCHO at 300 K (Chance and Orphal, 2011) |
| | $O_3$ at 228 K and 295 K (Malicet et al., 1995; Daumont et al., 1992) |
| | $NO_2$ at 220 K (Vandaele et al., 1998)[b] |
| | BrO at 228 K (Wilmouth et al., 1999) |
| | $O_4$ at 293 K (Thalman and Volkamer, 2013)[b] |
| Ring effect | Chance and Spurr (1997)[b] |
| Common mode | On-line common mode from easternmost swaths (143-150°E) for a day |
| Scaling and baseline polynomials | 3rd order |
| **AMF look-up table parameters** | |
| Longitude (degree) (n=33) | 70 to 150 with 2.5 grid |
| Latitude (degree) (n=30) | -4 to 54 with 2.0 grid |
| Solar Zenith Angle (degree) (n=9) | 0, 10, 20, 30, 40, 50, 60, 70, 80 |
| Viewing Zenith Angle (degree) (n=9) | 0, 10, 20, 30, 40, 50, 60, 70, 80 |
| Relative Azimuth Angle (degree) (n=3) | 0, 90, 180 |
| Cloud Top Pressure (hPa) (n=7) | 900, 800, 700, 600, 500, 300, 100 |
| Surface Albedo (n=7) | 0, 0.1, 0.2, 0.3, 0.4, 0.6, 0.8, 1.0 |

[a] GEMS fitting parameters follow González Abad et al. (2015). However, undersampling is not included in the fitting parameters for GEMS, and reference sectors for radiance reference and common mode are different.

[b] The datasets are used in QA4ECV retrievals. Please refer to De Smedt et al. (2018) for other datasets and fitting options.

How will the cloud retrieval from GEMS work? What surface reflectivity data will be used for the cloud and HCHO retrievals? How does the GEMS team address the issue of viewing geometry dependent surface reflectivity? These issues are not discussed, and thus the paper runs the risk of being read as just another OMI HCHO approach, i.e. of little specificity to GEMS.

*Thanks for your comments. Effective cloud fraction and cloud top pressure (effective*

*cloud pressure) from GEMS will be retrieved by using O₄ absorption band with the assumption of Lambertian surface reflectors. Surface reflectance is provided as Lambertian equivalent reflectivity (LER) from GEMS Level 2 surface properties and is used for cloud and HCHO retrievals. GEMS LER products are retrieved as composites of minimum LER values for 15 days every hour with fixed viewing geometry so that geometry dependent LER is yielded.*

*We briefly referred to input parameters provided from GEMS Level 2 for AMF calculation and added references as follows:*

Surface albedo, effective cloud fraction, and cloud top pressure are retrieved from GEMS and are used in the AMF calculations. GEMS Level 2 surface properties include Lambertian equivalent reflectivity (LER) and the daily bidirectional reflectance distribution function (BRDF) (Lee and Yoo, 2018). GEMS LER products are retrieved as composites of minimum LER values for 15 days every hour with fixed viewing geometry so that geometry dependent LER are yielded. The effective cloud fraction and cloud top pressure (effective cloud pressure) are retrieved from GEMS with the assumption of a Lambertian cloud surface (cloud surface albedo = 0.8) (Veefkind et al., 2016). GEMS surface reflectivity products are also used for cloud retrievals. In addition, the radiative cloud fraction ($f_{rc}$) will be provided from GEMS Level 2 cloud products, and is defined by Eq. 9, where $I_{cld}$ and $I_{clr}$ are radiances over cloud and cloud-free surfaces, respectively.

3. The paper would be strengthened if the authors would provide a breakup of the uncertainty budget for typical polluted and clean conditions, e.g. in the form of a table.

*We summarized the uncertainties of HCHO VCD due to AMF uncertainties for polluted and clean regions in Table 2 and discussed it as follows:*

Table 2 summarizes estimated retrieval uncertainties of GEMS HCHO VCDs due to AMF uncertainties as functions of surface albedos, cloud top pressures, and cloud fractions. Values are calculated assuming conditions with solar zenith angle of 30°, viewing zenith

angle of 30°, relative azimuth angle of 0°, cloud fractions less than 0.3, and a profile height of 700 hPa. Uncertainties of HCHO VCDs due to AMF uncertainties can be as large as 20% and 24% of HCHO VCDs in clean and polluted areas, respectively. Maximum values occur for conditions with low surface albedo and clouds at high altitudes, and high cloud fractions, but they do not differ much between clean and polluted areas. However, AMF driven HCHO uncertainty with respect to the profile height in polluted areas is higher than that in clean areas, implying that accurate HCHO profile information in polluted areas is important for the GEMS HCHO retrieval. We can minimize the a priori HCHO profile uncertainties by using averaging kernels.

**Table 2. Retrieval uncertainties of GEMS HCHO VCD due to AMF uncertainties as functions of surface albedos, cloud top pressures, cloud fractions, and HCHO profile heights for clean and polluted areas. Values are calculated for conditions with solar zenith angle of 30°, viewing zenith angle of 30°, relative azimuth angle of 0°, cloud fractions less than 0.3, and a profile height of 700 hPa.**

| AMF contribution to HCHO VCD uncertainty | Clean | Polluted |
|---|---|---|
| Surface albedo ($\alpha_s$) | 1-10% | 1-12% |
| Cloud top pressure ($p_c$) | 0-11% | 0-11% |
| Cloud fraction ($f_c$) | 0-19% | 0-17% |
| HCHO height ($p_h$) | 0-11% | 0-17% |
| Total | 2-20% | 3-24% |

4. I'm missing a discussion of the GEOS-Chem 2x2.5 a priori profile shapes. These are much coarser than the 7x8 km2 viewing scenes, and this will result in a substantial AMF uncertainty. It is true that this can be accounted for via application of the averaging kernels, or by recomputing the AMFs with high-resolution profiles from a regional CTM or zoom-version of GEOS-Chem. In any case this issue should be discussed in more detail, and also included in the uncertainty budget.

*We defined a profile height parameter ($p_h$) as an altitude below which 75% of HCHO VCDs exist from the surface to estimate AMF uncertainty with respect to a HCHO*

*profile shape. We included AMF uncertainty with respect to a profile height. We discussed as follows:*

The AMF uncertainty can be estimated by each parameter in Eq. 16. We examine AMF uncertainties for surface albedo ($\alpha_s$), cloud top pressure ($p_c$), and effective cloud fraction ($f_c$) with a solar zenith angle of 30°, a viewing zenith angle of 30°, and a relative azimuth angle of 0°. In addition, we define a profile height parameter ($p_h$) as an altitude below which 75% of HCHO VCDs exist from the surface, to estimate AMF uncertainty with respect to a HCHO profile shape. The uncertainties of parameters ($\sigma_{\alpha_s} = 0.02$, $\sigma_{p_c} = 50\ hPa$, and $\sigma_{f_c} = 0.05$) are based on De Smedt et al. (2018) and will be replaced to those from GEMS Level 2 products. The uncertainty of profile height ($\sigma_{p_h}$) is defined as a standard deviation of profile heights in AMF LUT, and $\sigma_{p_h}$ in polluted and clean areas are 84 and 55 hPa, respectively.

$$\sigma_{AMF}^2 = \left(\frac{\partial AMF}{\partial \alpha_s}\sigma_{\alpha_s}\right)^2 + \left(\frac{\partial AMF}{\partial p_c}\sigma_{p_c}\right)^2 + \left(\frac{\partial AMF}{\partial f_c}\sigma_{f_c}\right)^2 + \left(\frac{\partial AMF}{\partial p_h}\sigma_{p_h}\right)^2 \qquad (16)$$

…

Figure 5d shows increasing AMF values with an increase in the profile height, resulting from increased HCHO absorptions at high altitudes. The AMF sensitivity to profile heights in clean areas is higher than that in polluted areas because HCHO distributions are more uniform in clean areas than polluted areas.

[Figure]

**Figure 5. AMF variations as functions of (a) surface albedo, (b) cloud top pressure (CTP), (c) effective cloud fraction ($f_c$), and (d) profile height over clean (blue) and polluted (red) areas. Conditions of the AMF LUT are given in the figures. For sensitivity to surface albedo, cloud-free conditions are assumed. For sensitivity to cloud fraction, cloud top pressures are 800 hPa (solid line) and 500 hPa (dashed line).**

5. It remains very much unclear how the latitude-bias is being determined. The text on page 11 (lines 9-11) is not clear, and the patterns shown in Figure 5(d) need explanation.

*Latitudinal bias is determined by retrieved slant columns for radiance references.*

*Figure 5d (Figure 6d in the new manuscript) shows latitudinal bias, which is equal to averaged slant columns for radiance references as a function of latitude.*
*We modified sentences as follows:*

In addition, we need to correct latitudinal biases for OMI. Previous studies explained that the latitudinal biases result from spectral interferences of BrO and $O_3$, whose concentrations are a function of latitude and are high in high latitudes (De Smedt et al., 2008; De Smedt et al., 2015; González Abad et al., 2015). Therefore, the latitudinal biases were corrected when a radiance reference was used as the reference spectrum (De Smedt et al., 2008; González Abad et al., 2015; De Smedt et al., 2018). We correct the latitudinal biases, which are slant columns retrieved for a radiance reference and are averaged as a function of latitude, by subtracting the biases from the corrected slant columns in Eq. 11.
…
Figure 6d shows the absolute differences between OMI HCHO slant columns with and without latitudinal bias corrections from the GEMS algorithm as latitudinal biases. Slant columns with bias corrections increase at latitudes lower than 5°N and higher than 25°N but decrease at latitudes from 5°N-25°N.

Minor comments

P2, L12: suggest to remove 'instrument' after SCIAMACHY.
*We removed it.*

P3, L15: suggest to use air quality in the singular
*We changed the in the singular.*

P3, L16-17: compared to TROPOMI's 7x7 km2 pixels, the 7x8 km2 resolution from GEMS is not that superior, so I suggest to nuance that statement.
*We modified the sentence as follows:*
Instruments on-board these geostationary satellites have good spatial resolutions corresponding to those of TROPOMI and high signal-to-noise ratios, …

P6: eq. (3) and (4) – suggest to use that mathematical e rather than exp which reads as computer code.

*We changed "exp" into "e" in other equations (Eq. (5) and (6)) as well as Eq. (3) and (4).*

Eq. (15) appears wrong.

*We corrected Eq. 15 as follows:*

$$\sigma_{s,j}^2 = RMS^2 \frac{m}{m-n} C_{j,j} C_{j,j},$$ (15)

Figure 7: what are the relative uncertainties in the AMF?

*We deleted Figure 7 but add Table 2 to describe retrieval uncertainties of GEMS HCHO VCDs due to AMF uncertainties. We discussed Table 2 above.*

Validation: which spatio-temporal selection criteria were used?

*For comparison with OMI other products, we used monthly averages weighted by fitting uncertainties and overlapped areas between pixels and grid boxes with a horizontal resolution of 0.25° × 0.25°.*

*For MAX-DOAS comparison, we also used the weighted monthly averages for OMI in a grid box of 0.25° at the center of site locations, and MAX-DOAS data were arithmetically averaged within OMI overpass time (12:00-15:00 local time). We updated comparisons between MAX-DOAS and OMI products for a year at OHP and Bremen in 2005 and at Xianghe in 2016.*

*We modified paragraphs as follows:*

We also compare satellite results with MAX-DOAS ground observations at Haute-Provence Observatory (OHP) in France, Bremen in Germany, and Xianghe in China (Table 4). MAX-DOAS data are collected within the OMI overpass time (12:00–15:00

local time) at OHP and Bremen in 2005, and at Xianghe in 2016, respectively. We collect OMI data pixels that are overlapped by a grid box of 0.25° at the center of the site location, and average values of OMI data are weighted by uncertainties and overlapped areas between pixels and grid boxes.

Comparisons of HCHO VCDs between MAX-DOAS and satellite products are shown in Fig. 11 and Table 4. Averaged MAX-DOAS HCHO VCDs for a year are $7.6 \times 10^{15}$, $6.7 \times 10^{15}$, and $1.6 \times 10^{16}$ molecules cm$^{-2}$ at OHP, Bremen, and Xianghe, respectively. HCHO VCDs show a seasonal variation with the maximum concentrations in summer at all sites (Fig. S3). The largest monthly change is shown at Xianghe, likely driven by abundant VOC precursors for HCHO productions compared to OHP and Bremen.

Averaged HCHO VCDs from OMI GEMS are by 16%, 9%, 25% lower than those from MAX-DOAS at OHP, Bremen, and Xianghe. At Bremen, HCHO VCDs from the GEMS algorithm are in the best agreement with those of MAX-DOAS and show similar monthly variations with MAX-DOAS. OMI GEMS results at Xianghe show a monthly variation but at OHP do not show monthly variation despite of a bit increment in summer. In particular, the GEMS algorithm yields lower HCHO VCDs in summer. These lower values may be caused by the a priori HCHO profiles used in AMF calculation. In summer, HCHO is produced and concentrated near the surface, which results in lower AMFs (higher VCDs). S. W. Kim et al. (2018) showed the anti-correlation between AMF values and the HCHO mixing ratios at 200 m above ground level. OMHCHO products show similar tendencies as OMI GEMS, but they are much lower than those of OMI GEMS. OMI QA4ECV products are higher than MAX-DOAS at OHP and Bremen but are in the best agreement with MAX-DOAS at Xianghe compared to other satellite products.

**Table 4. Averaged HCHO VCDs (molecules cm$^{-2}$) from MAX-DOAS ground observations and OMI satellite data at OHP in France, Bremen in Germany, and Xianghe in China. For satellites, mean values are weighted by uncertainties and overlapped areas between satellite pixels and 0.25° grid cells for each site. Relative differences between OMI and MAX-DOAS are given in parentheses.**

| Site[a] | Class[b] | MAX-DOAS[c] | OMHCHO | OMI QA4ECV | OMI GEMS |
|---------|----------|-------------|--------|------------|----------|

| | | | | | |
|---|---|---|---|---|---|
| OHP (44°N, 5.5°E) | Rural | $7.5 \times 10^{15}$ | $5.8 \times 10^{15}$ (-24%) | $1.1 \times 10^{16}$ (50%) | $6.3 \times 10^{15}$ (-16%) |
| Bremen (53°N, 9°E) | Urban | $6.7 \times 10^{15}$ | $5.1 \times 10^{15}$ (-23%) | $9.3 \times 10^{15}$ (40%) | $6.1 \times 10^{15}$ (-9%) |
| Xianghe (39°N, 117°E) | Sub-urban | $1.6 \times 10^{16}$ | $1.0 \times 10^{16}$ (-37%) | $1.7 \times 10^{16}$ (4%) | $1.2 \times 10^{16}$ (-25%) |

[a] HCHO VCDs are averaged at OHP and Bremen in 2005 and at Xianghe in 2016.

[b] Class is assigned in a QA4ECV MAXDOAS website (http://uv-vis.aeronomie.be/groundbased/QA4ECV_MAXDOAS)

[c] Fitting windows of 336–359 nm and 324–359 nm are used at OHP and Bremen, and at Xianghe, respectively.

[Figure]

**Figure 11. HCHO vertical columns from MAX-DOAS, OMHCHO, OMI QA4ECV, and OMI GEMS at OHP and Bremen in 2005, and at Xianghe in 2016. Orange lines are median values for each dataset, and blue diamonds are mean values. We computed mean values of each satellite product weighted by uncertainties and overlapped areas between satellite pixels and 0.25° grid cells for each site. Boundaries of boxes indicate first and last quantiles of datasets.**

---

## Author Comment (AC3) · 15 May 2019

**Responses to Referee's Comments**

*We appreciate careful reading and lots of valuable comments.*
*We wrote referee's comments in black, our responses to comments in blue and italics,*
5 *and the revised manuscript in red.*

**Referee #3:**

**General Description:**

The authors describe the retrieval algorithm of formaldehyde (HCHO) for the future
10 GEMS instrument and estimate the likely uncertainties and biases relative to OMI and
ground-based MAX-DOAS measurements. The content is appropriate for AMT.
Suggested changes, comments and concerns are included below.

**General Comments:**

15 It's not clear what's unique about the retrieval to GEMS. Seems more like a recapitulation
of the OMI retrieval description paper of González Abad et al. (2015). A way to address
this would be to assess the implication of the unique temporal component of GEMS (i.e.
observations throughout the day) on uncertainties in the retrieval.

20 *Thanks for suggestions. We analyzed expected random uncertainty for GEMS by using*
*simulated radiances, which are convoluted with GEMS bandpass functions at 330 nm*
*and include noises based on signal-to-noise ratio for co-added pixels with spatial*
*resolutions of $7 \times 8$ km$^2$. We updated related paragraphs as follows:*

25 We analyze expected uncertainties for the GEMS algorithm by using simulated radiances
from Kwon et al. (2017) and OMI Level 1B data. In order to estimate the expected random
uncertainty for GEMS (Section 3.1.1), we use simulated radiances, which are convoluted
with GEMS bandpass functions at 330 nm as a function of cross-track positions in the
south to north direction. Simulated radiances include noises based on the expected signal-
30 to-noise ratio for co-added pixels with spatial resolutions of $7 \times 8$ km$^2$. We use absorption
cross-sections of Ring effect, $O_3$, $NO_2$, HCHO, and additionally $SO_2$ (Hermans et al.,

2009; Vandaele et al., 2009) in radiance fitting because $O_3$, $NO_2$, and HCHO, and $SO_2$ were considered in radiance calculation (Kwon et al. 2017).

For other uncertainty analyses, we use OMI Level 1B data with OMI slit function data (Dirksen et al., 2006) in order to examine algorithm sensitivities to individual parameters. Fitting options such as absorption cross-section data and the fitting window are summarized in Table 1. It will be necessary to conduct an additional uncertainty analysis for GEMS HCHO retrievals after GEMS is launched.

…

Random uncertainties from the GEMS algorithm are estimated using simulated radiances. RMS of fitting residuals and random uncertainty for the GEMS domain range from $2.9 \times 10^{-4}$ to $2.1 \times 10^{-3}$ and $2.1 \times 10^{15}$ to $1.6 \times 10^{16}$ molecules cm$^{-2}$, respectively, which are comparable with those (RMS: $4 \times 10^{-4}$ to $2.0 \times 10^{-3}$; random uncertainty: $3.3 \times 10^{15}$ to $1.8 \times 10^{16}$ molecules cm$^{-2}$) obtained from the GEMS algorithm using OMI Level 1B data. GEMS measures target species every hour in daytime so that changes of solar location for a day can affect the accuracy of radiance fitting. An averaged fitting RMS value and a random uncertainty are $6.9 \times 10^{-4}$ and $5.0 \times 10^{15}$ molecules cm$^{-2}$ for conditions with both solar and viewing zenith angles less than 70, which happen at 8:00–18:00 and 9:00–16:00 local time of Seoul in summer and winter, respectively. However, the fitting RMS value and the random uncertainty increase to $1.1 \times 10^{-3}$ and $8.2 \times 10^{15}$ molecules cm$^{-2}$, respectively, when solar and viewing zenith angles are higher than 70.

*To clarify, we remained descriptions related with GEMS in the Section 2. Descriptions related with OMI to validate the GEMS algorithm were moved to new Section 4.1.*

*We described a radiance reference for GEMS in Section 2.2.3 as follows:*

Table 1 summarizes the detailed information used in the GEMS HCHO retrieval algorithm. We follow fitting options in González Abad et al. (2015). We use measured

radiances as the reference spectrum, called a radiance reference, and measured radiances are averaged over the easternmost swaths (143-150°E; shaded areas in Fig. 1) for a day as a function of cross-track positions in the south to north direction. Background corrections are required when we use a radiance reference and are discussed in Section 2.2.5. Also, GEMS has cross-track swaths in the south to north directions while instruments such as OMI and TROPOMI have west to east swath. Therefore, latitudinal biases resulting from BrO and $O_3$ latitude-dependent interferences can be minimized for GEMS and are discussed in Section 4.1.

*We described GEMS surface reflectivity and cloud information used for AMF calculation in Section 2.2.4.*

Surface albedo, effective cloud fraction, and cloud top pressure are retrieved from GEMS and are used in the AMF calculations. GEMS Level 2 surface properties include Lambertian equivalent reflectivity (LER) and the daily bidirectional reflectance distribution function (BRDF) (Lee and Yoo, 2018). GEMS LER products are retrieved as composites of minimum LER values for 15 days every hour with fixed viewing geometry so that geometry dependent LER are yielded. The effective cloud fraction and cloud top pressure (effective cloud pressure) are retrieved from GEMS with the assumption of a Lambertian cloud surface (cloud surface albedo = 0.8) (Veefkind et al., 2016). GEMS surface reflectivity products are also used for cloud retrievals. In addition, the radiative cloud fraction ($f_{rc}$) will be provided from GEMS Level 2 cloud products, and is defined by Eq. 9, where $I_{cld}$ and $I_{clr}$ are radiances over cloud and cloud-free surfaces, respectively.

*Also, we wrote a plan to consider temporal variations of a priori HCHO profiles as follows:*

However, the horizontal resolution of 2° × 2.5° for HCHO profiles in AMF LUT is much coarser than the GEMS horizontal resolution of 7 × 8 km² to discern spatial variations by

local source emissions. HCHO profiles in AMF LUT are monthly averaged so that hourly variations are not accounted for. In order to resolve these rough conditions, we can use HCHO profiles with a finer resolution as a function of time. For example, Kwon et al. (2017) showed that HCHO retrievals using monthly mean hourly AMF values were in better agreement with the model simulations in observation system simulation experiments (OSSE) than those using monthly mean AMF values. Also, air quality forecasting data can be used to consider hourly varying HCHO profiles. Further studies are required to examine the dependency of AMF calculations on spatial resolutions and temporal variations of HCHO profiles and its effect on GEMS retrieval.

Throughout, use the standard symbol $\otimes$ for convolution. This will help clarify terms in equations that are confusing, as brackets are used to denote dependence, but also operators, e.g. $f \otimes g(\lambda)$ to replace $(f * g)(\lambda)$ in Equation (2) is clearer. Please correct these issues throughout.

*Thanks for suggestion. We replaced the symbol * to the symbol $\otimes$ in Eq. (1)-(5) as follows:*

$$I_R(\lambda) = I_0^h \otimes g(\lambda + \Delta\lambda)P_{sc}(\lambda) + P_{bl}(\lambda), \tag{1}$$

$$f \otimes g(\lambda) = \int_{-\infty}^{\infty} f(\Lambda)g(\lambda - \Lambda)d\Lambda \tag{2}$$

$$\text{attenuated radiance in radiance fitting} = I_0^h \otimes g(\lambda)e^{-\tau^h \otimes g(\lambda)}, \tag{3}$$

$$\text{attenuated radiance in reality} = (I_0^h(\lambda)e^{-\tau^h(\lambda)}) \otimes g(\lambda). \tag{4}$$

$$\sigma_{ps}(\lambda) = \frac{1}{scd_{ref}}\ln\left(\frac{I_0^h \otimes g(\lambda)}{\left(I_0^h(\lambda)e^{-scd_{ref}\sigma^h(\lambda)}\right) \otimes g(\lambda)}\right), \tag{5}$$

Inconsistent use of wavelength dependence in equations. For example, why do $I_R$ and $I_0^h$ not depend on wavelength in Equations (1)-(4), but do in Equation (5)?

*We changed those equations above, and we also modified variables related with Eq. (6) as a function of wavelength.*

$$I(\lambda) = \left[\left(aI_0(\lambda) + c_r\sigma_r(\lambda)\right)e^{-\sum_i SCD_i\sigma_i(\lambda)} + c_{cm}\sigma_{cm}(\lambda)\right]P_{sc}(\lambda) + P_{bl}(\lambda), \qquad (6)$$

Many sub-sections in Section 2.2. are the same as in González Abad et al. (2015). Why not just refer the reader to that paper and only state aspects specific to GEMS and that are different between the two approaches?

*As we answered to first comments, we remained descriptions related to GEMS. We also added new sub-section 4.1 to describe fitting options in the GEMS algorithm for OMI HCHO retrievals.*

**4.1 Retrieval of OMI HCHO**

GEMS fitting options described in Table 1 are largely consistent with those of OMHCHO products (González Abad et al., 2015). However, we do not include spectral undersampling (Chance et al., 2005) in the fitting process for GEMS, and reference sectors for a radiance reference are 143-150°E (shaded areas in Fig. 1). For OMI products, spectral undersampling needs to be included, and radiance references are from the Pacific Ocean as described in González Abad et al. (2015). We use simulated HCHO vertical columns for the background correction, which are zonally and monthly averaged over the reference sector (140-160°W, 90°S-90°N) except for Hawaii (154-160°W, 19-22°N).

In addition, we need to correct latitudinal biases for OMI. Previous studies explained that the latitudinal biases result from spectral interferences of BrO and $O_3$, whose concentrations are a function of latitude and are high in high latitudes (De Smedt et al., 2008; De Smedt et al., 2015; González Abad et al., 2015). Therefore, the latitudinal biases were corrected when a radiance reference was used as the reference spectrum (De Smedt et al., 2008; González Abad et al., 2015; De Smedt et al., 2018). We correct the latitudinal biases, which are slant columns retrieved for a radiance reference and are averaged as a function of latitude, by subtracting the biases from the corrected slant columns in Eq. 11. Figure 6 shows OMI HCHO slant columns from OMHCHO products (Fig. 6a) and the GEMS algorithm without and with latitudinal bias corrections (Fig. 6b and 6c). HCHO slant columns without latitudinal bias corrections (Fig. 6b) are retrieved larger in 5°N-25°N than OMHCHO products, but HCHO slant columns with the bias corrections are in

better agreement with OMHCHO products. Figure 6d shows the absolute differences between OMI HCHO slant columns with and without latitudinal bias corrections from the GEMS algorithm as latitudinal biases. Slant columns with bias corrections increase at latitudes lower than 5°N and higher than 25°N but decrease at latitudes from 5°N-25°N. However, latitudinal biases can be minimized when using a radiance reference as a function of each cross-track position in the south to north direction for GEMS. In default fitting options, therefore, we do not include latitudinal correction and do not analyze uncertainty of latitudinal corrections in Section 3. However, a further investigation for the latitudinal biases needs to be required after GEMS is launched.

Figure 7 shows an example of retrieved HCHO optical depths and fitting residuals as functions of wavelengths for a pixel in Indonesia (March 23 2005; orbit 3655). The retrieved HCHO slant column is $3.2 \times 10^{16}$ molecules $cm^{-2}$, which is relatively high due to biomass burning in that region. Average slant column and random uncertainty for all pixels on the orbit are $7.6 \times 10^{15}$ and $6.9 \times 10^{15}$ molecules $cm^{-2}$, respectively, over the GEMS domain. The large random uncertainty of 100% or larger results from pixels with low concentrations, where averaged slant columns and random uncertainties are $2.2 \times 10^{15}$ and $6.2 \times 10^{15}$ molecules $cm^{-2}$.

It's not clear why Section 2.2.5 is relevant, as it describes bias corrections specific to OMI. Is it anticipated that the same bias corrections will be needed for GEMS? If this section is relevant, the readers could just be referred to González Abad et al. (2015) and this section be kept brief.

*Thanks for your comments. For GEMS, background corrections are only used when we use a radiance reference. To clarify, therefore, we explained background corrections for GEMS in Section 2.2.5, and corrections and discussions for OMI were moved to Section 4.1.*

*We modified paragraphs in Section 2.2.5 as follows:*

An alternative method to avoid the above-mentioned biases in the fitting procedure is to

use measured radiances over a clean background region (referred to as radiance references) as the reference spectrum in radiance fitting. As measured radiance includes instrument noise and attenuation by interfering gases in the background atmosphere, the interfering effects can be minimized in radiance fitting, which results in negligible cross-track biases.

For GEMS, we plan to use simulated HCHO columns over easternmost regions (143-150°E) as GEMS reference sectors, which are shaded areas in Fig. 1. The GEMS reference sectors include part of islands near the equator and Japan but are relatively clean areas in south/north direction over the GEMS domain. In comparisons with background HCHO vertical columns over the Pacific Ocean for OMI (Fig. S1), annual mean of GEMS background columns over 4°S–45°N is $3.3 \times 10^{15}$ molecules cm$^{-2}$ slightly higher than that of OMI background columns ($3.2 \times 10^{15}$ molecules cm$^{-2}$), showing that we can use easternmost regions as background in the GEMS domain. Occasionally, local differences between GEMS and OMI background columns can be as large as $3.8 \times 10^{15}$ molecules cm$^{-2}$ in the tropical region of the southern hemisphere due to biogenic activity and biomass burning, but the standard deviation of background values in that region is $5.1 \times 10^{14}$ molecules cm$^{-2}$ even lower than that of $1.2 \times 10^{15}$ molecules cm$^{-2}$ in the middle latitude (>30°N), indicating that the influences from biogenic activity and biomass burning can be corrected by model simulations.

The retrieved slant columns using a radiance reference are differential slant columns ($\Delta SCD = SCD - SCD_0$) and do not include background HCHO columns ($SCD_0$) that are mainly from the oxidation of methane. To account for the background columns, we use HCHO vertical columns simulated in 2014 from a chemical transport model, GEOS-Chem (Bey et al., 2001) with a spatial resolution of 2° × 2.5°. Simulated HCHO vertical columns are zonally and monthly averaged over the reference sectors and are interpolated to 720 latitudinal grid points with a resolution of 0.25° from 90°S to 90°N.

In order to account for dependency of measured radiances on geometric angles, we then convert simulated background vertical columns into slant columns by applying AMF values over the reference sector ($AMF_0$), which are calculated with cloud information and geometric angles on the reference sectors. Corrected GEMS HCHO slant columns are formulated as the sum of the retrieved differential slant columns and the simulated background slant columns as shown in Eq. 11,

$$\Omega_s(i,j) = SCD_{corr}(i,j) = \Delta SCD(i,j) + AMF_0(lat)VCD_m(lat), \qquad (11)$$

where i and j indicate pixel indices of cross and along tracks, respectively, and $VCD_m$ denotes a background vertical column density from the model. We finally apply AMF values from the LUT to the corrected slant columns to obtain GEMS HCHO vertical column densities.

It's also not clear why data quality flags are provided for a future product. This would only be important for the user when the data is ready for release.

*The data quality flag is provided for basic information of data quality in radiance fitting, and we followed the flag definition from González Abad et al. (2015). We are planning to provide flags including much information such as geometry angles, clouds, surface information.*

Section 3 appears to just be testing uncertainties inherent in fitting parameters and retrieval terms that would be an issue for all space-based instruments measuring HCHO, rather than being specific to GEMS. Is there anything unique to GEMS (instrument configuration, viewing domain, repeat time etc.) that would increase or decrease sensitivity to these uncertainties relative to other instruments?

*Uncertainty related to GEMS instrument is considered in random uncertainty. Random uncertainty, called fitting uncertainty, is calculated from fitting residuals caused by instrument noise, radiance measurement uncertainty from dark current and stray lights, and polarization. We estimated expected random uncertainty by using simulated radiances with GEMS bandpass functions and signal-to-noise ratios. We discussed it in the first answer.*

*GEMS does not include a polarization scrambler while OMI and TROPOMI include a polarization scrambler. In the operation, polarization correction will be conducted when L1B data are produced. The correction could minimize polarization, but it would*

*not be perfect. The effects could increase random uncertainty. We need to have a process to minimize polarization.*

*We discussed it in Section 5 as follows:*

5 We currently use a broad fitting window (328.5–356.0 nm). However, we may need to use a different fitting window to reduce interference from polarization effects because GEMS does not include a polarization scrambler. A polarization correction is planned to minimize its interference during GEMS Level 1B production, but we need to examine the retrieval sensitivity to polarization.

**Specific Comments:**

P2, Line 18: the spatial resolution of TROPOMI is finer than 7 x 7 km2 for HCHO (De Smedt et al., 2018).

15

*Veefkind et al. (2012) showed the spatial resolution of TROPOMI UVIS band 3 (310-405 nm) is 7 x 7 km². However, we found TROPOMI HCHO products are provided with 7 x 3.5 km². We corrected it from 7 x 7 km² to 7 x 3.5 km².*

20

Equation (1): Why are *Psc* and *Pbl* not dependent on wavelength?

*$P_{sc}$ and $P_{bl}$ are functions of wavelength. Therefore, we changed it as follows:*

25 $$I_R(\lambda) = I_0^h \otimes g(\lambda + \Delta\lambda)P_{sc}(\lambda) + P_{bl}(\lambda), \tag{1}$$

P3, Line 22: Can aerosol optical properties be retrieved across this wavelength and for this type of instrument? Do the authors mean AOD and aerosol index (AI)?

30

*We meant AOD and SSA. We clarified it as follows:*

Geostationary Environment Monitoring Spectrometer (GEMS) will be launched by South Korea, and it will measure radiances ranging from 300 to 500 nm every hour with fine spatial resolutions of $3.5 \times 8$ km$^2$ for aerosols or $7 \times 8$ km$^2$ for gases over Seoul in South Korea to monitor column concentrations of air pollutants including $O_3$, $NO_2$, $SO_2$, and HCHO, and aerosol optical properties (aerosol optical depth and single scattering albedo).

Table 1: add references for these parameters as footnotes to point to consistency with existing retrievals.

*We marked 'a' and 'b' on datasets used in OMHCHO and QA4ECV, respectively, and explanation was written in footnotes as follows:*

| Radiance fitting parameters[a] | |
|---|---|
| Fitting window (calibration window) | 328.5–356.5 nm (325.5–358.5 nm) |
| Radiance reference | Measured radiances from far east swaths (143-150°E) for a day |
| Solar reference spectrum | Chance and Kurucz (2010)[b] |
| Absorption cross-sections | HCHO at 300 K (Chance and Orphal, 2011) |
| | $O_3$ at 228 K and 295 K (Malicet et al., 1995; Daumont et al., 1992) |
| | $NO_2$ at 220 K (Vandaele et al., 1998)[b] |
| | BrO at 228 K (Wilmouth et al., 1999) |
| | $O_4$ at 293 K (Thalman and Volkamer, 2013)[b] |
| Ring effect | Chance and Spurr (1997)[b] |
| Common mode | On-line common mode from easternmost swaths (143-150°E) for a day |
| Scaling and baseline polynomials | 3$^{rd}$ order |

[a] GEMS fitting parameters follow González Abad et al. (2015). However, undersampling is not included in the fitting parameters for GEMS, and reference sectors for radiance reference and common mode are different.

[b] The datasets are used in QA4ECV retrievals. Please refer to De Smedt et al. (2018) for other datasets and fitting options.

P8, Lines 14-17: What about clouds (Millet et al., 2006)?

*We corrected the sentence as follows:*

AMF uncertainties contribute to retrieval uncertainties by multiple factors including cloud, HCHO vertical distribution, aerosol vertical distribution, and aerosol optical properties (Millet et al., 2006; Chimot et al., 2016; Kwon et al., 2017; Hewson et al., 2015).

P18, Lines 8-14: Comment too on the implications of more observations over the same scene per day on uncertainty compared to OMI.

*Thank your comments. We wanted to show an example of OMI HCHO results retrieved*
15 *from the GEMS algorithm. Therefore, we showed HCHO optical depths and fitting residuals and explained averaged HCHO slant column density and random uncertainty. In addition, we explained slant columns and random uncertainties in pixels with low concentrations as follows:*

20 Averaged slant column and random uncertainty for all pixels on the orbit are $7.6 \times 10^{15}$ and $6.9 \times 10^{15}$ molecules cm$^{-2}$, respectively, over the GEMS domain. The large random uncertainty of 100% or larger results from pixels with low concentrations, where averaged slant columns and random uncertainties are $2.2 \times 10^{15}$ and $6.2 \times 10^{15}$ molecules cm$^{-2}$.

25

P18, Lines 26-28: Provide an appropriate reference for this statement.

*We added a reference as follow:*
Zhong, L., Louie, P. K. K., Zheng, J., Yuan, Z., Yue, D., Ho, J. W. K., and Lau, A. K. H.:
30 Science–policy interplay: Air quality management in the Pearl River Delta region and Hong Kong, Atmospheric Environment, 76, 3-10,

https://doi.org/10.1016/j.atmosenv.2013.03.012, 2013

Referencing: some references are missing the doi number (e.g., González Abad et al., 2015).

*DOI numbers are added as follows:*

González Abad et al. (2015): 10.5194/amt-8-19-2015

Barkley et al. (2013): 118, 6849-6868, 10.1002/jgrd.50552, 2013

10    Bey et al. (2001): 10.1029/2001JD000807

Cantrell et al. (1990): 10.1021/j100373a008

Chance et al. (1997): 10.1364/AO.36.005224

Chance et al. (2000): 10.1029/2000GL011857

Daumont et al. (1992): 10.1007/BF00053756

15    De Smedt et al. (2008): 10.5194/acp-8-4947-2008

Hewson et al. (2013): 10.5194/amt-6-371-2013

Malicet et al. (1995): 10.1007/BF00696758

Marais et al. (2012): 10.5194/acp-12-6219-2012

Palmer et al. (2001): 10.1029/2000JD900772

20    Spurr (2006): 10.1016/j.jqsrt.2006.05.005

Zhu et al. (2014): 10.1088/1748-9326/9/11/114004

**References:**

25    González Abad et al., Atmos. Meas. Tech., 8, 19-32, 2015, doi:10.5194/amt-8-19-2015.

De Smedt et al., Atmos. Meas. Tech., 11, 2395–2426, 2018, doi:10.5194/amt-11-2395-2018. Millet et al., J. Geophys. Res., doi:10.1029/2005JD006853, 2006

---

## Author Comment (AC4) · 15 May 2019

[Figure]

**Figure S3. Monthly variations of HCHO VCDs from MAX-DOAS (black) and OMI at OHP, Bremen, and Xianghe. Blue indicates OMHCHO, red indicates QA4ECV, and green indicates GEMS.**